# Oxidized nucleotide insertion by pol β confounds ligation during base excision repair

Melike Çağlayan[1], Julie K. Horton[1], Da-Peng Dai[1], Donna F. Stefanick[1] & Samuel H. Wilson[1]

Oxidative stress in cells can lead to accumulation of reactive oxygen species and oxidation of DNA precursors. Oxidized purine nucleotides can be inserted into DNA during replication and repair. The main pathway for correcting oxidized bases in DNA is base excision repair (BER), and in vertebrates DNA polymerase β (pol β) provides gap filling and tailoring functions. Here we report that the DNA ligation step of BER is compromised after pol β insertion of oxidized purine nucleotides into the BER intermediate *in vitro*. These results suggest the possibility that BER mediated toxic strand breaks are produced in cells under oxidative stress conditions. We observe enhanced cytotoxicity in oxidizing-agent treated pol β expressing mouse fibroblasts, suggesting formation of DNA strand breaks under these treatment conditions. Increased cytotoxicity following MTH1 knockout or treatment with MTH1 inhibitor suggests the oxidation of precursor nucleotides.

[1] Genome Integrity and Structural Biology Laboratory, National Institutes of Health, National Institute of Environmental Health Sciences, Research Triangle Park, North Carolina 27709, USA. Correspondence and requests for materials should be addressed to S.H.W. (email: wilson5@niehs.nih.gov).

Oxidative stress is a common threat to genome stability[1]. In living cells, reactive oxygen species (ROS), such as superoxide and hydroxyl radicals, and reactive nitrogen species are formed continuously as a consequence of metabolic reactions, as well as exposure to environmental agents[2]. Cellular DNA and its precursor nucleotides are at risk of oxidization by ROS[3]. DNA oxidation leads to oxidized bases, abasic sites and strand breaks resulting in mutagenesis and cell death implicated in various age-related or neurodegenerative diseases[4]. A primary oxidative base modification in DNA is the oxidized form of guanine, 8-oxo-7,8-dihydro-2′-deoxyguanosine (8-oxodG)[5–7]. Furthermore, purine deoxynucleoside triphosphates (dNTPs) in the nucleotide pool are vulnerable to oxidation leading to oxidized forms of DNA precursors, that is, 7,8-dihydro-8′-oxo-dGTP (8-oxodGTP), 7,8-dihydro-8′-oxo-dATP (8-oxodATP) and 2-hydroxy-2′-deoxyadenosine-5′-triphosphate (2-OH-dATP)[8–10]. Oxidized dNTPs can be incorporated into DNA during DNA replication and repair resulting in deleterious effects[11–13]. To protect against these effects, oxidized bases are removed from the nucleotide pool by the sanitizing enzyme MutT Human Homolog 1 (MTH1) and from DNA by the base excision repair pathway (BER). DNA polymerase β (pol β) can incorporate 8-oxodGMP opposite either template base C or A, and is at risk of reinserting oxidized nucleotides during the gap-filling step of oxidized base BER[14–16].

BER, the predominant repair system for removal of ROS-induced lesions, involves a coordinated sequence enabling handoff of repair intermediates from one step to the next[17–19]. There is channelling of the repair intermediate after pol β nucleotide insertion to the final step in BER, DNA ligation, where a DNA ligase catalyzes phosphodiester bond formation between 3′-OH and 5′-P groups of the nicked BER intermediate[20,21]. Since DNA ligase activity requires a natural annealed base pair at the 3′-margin of the nick[22], mismatched or oxidized nucleotide insertion during the gap-filling step could result in disruption of BER[23]. However, the potential effect of oxidized base insertion on DNA ligation is not well understood. Our recent time-lapse crystallography observation with pol β indicated that post-insertion 3′-8-oxodGMP in a BER intermediate fails to remain base paired with either template base C or A and suggested the possibility DNA ligation could be compromised[16].

In the present study, we find that oxidized nucleotide insertion by pol β confounds the ligation step of BER, and results in abortive ligation or ligation failure, as revealed by formation of the 5′-adenylate product in vitro. The differences in ligation failure observed with pol β active site mutants point out a functional interaction between DNA ligase and pol β. We also find increased cytotoxicity of an oxidative stress-inducing agent in $pol\ \beta^{+/+}$ cells than in isogenic $pol\ \beta^{-/-}$ cells, and increased oxidative stress-induced cytotoxicity in $pol\ \beta^{+/+}MTH1^{-/-}$ cells as well as in the presence of a MTH1 inhibitor. Consistent with the idea of accumulation of strand breaks after ligation failure, higher levels of γH2AX staining are observed in oxidative agent-treated cells than in control untreated cells or in $pol\ \beta^{-/-}$ cells.

## Results

**Oxidized nucleotide insertion coupled with ligation failure.** To understand the effects of oxidized nucleotide insertion on the ligation step of BER, we initially measured pol β 8-oxodGMP insertion coupled with ligation in the same reaction mixture (Fig. 1a). Double FAM-labelled (at 5′- and 3′-ends) single-nucleotide gapped DNA substrates containing template bases A, T, C or G ($A^{gap}$, $T^{gap}$, $C^{gap}$ or $G^{gap}$, Supplementary Table 1) were used, and reactions were under conditions of DNA excess.

For the substrates $C^{gap}$ (Fig. 1c) and $A^{gap}$ (Fig. 1d), the observed ligation failure was associated with pol β-mediated 8-oxodGMP insertion and ligation over the time of incubation. For the substrate $C^{gap}$, while dGMP insertion along with ligation were detected at the initial time point (30 s; Fig. 1b, lane 2), weak insertion of 8-oxodGMP and negligible accumulation of the ligation product were observed at later times (Fig. 1b, lane 9). In this case, the 5′-adenylate product, that is AMP addition to the 5′-end of the substrate, also accumulated at later time points (Fig. 1b, lanes 11–15). In contrast, for correct dGTP insertion opposite template base C, more ligation products were observed over the time of incubation (Supplementary Fig. 1a). For the substrate $A^{gap}$, there was relatively strong accumulation of the 5′-adenylation product at earlier time points (10 s) with 8-oxodGTP (Fig. 1d).

An important consideration in interpretation of these results was that control experiments with DNA ligase I (Lig I) alone showed that ligation across the gap produced only a minimal amount of ligation product, as revealed by the difference in the size of the product (Supplementary Fig. 2a, compare lane 2 with lanes 3–12). In addition, 5′-adenylation was minimal in the absence of dNTP insertion and accumulated at very late time points of incubation (Supplementary Fig. 2b) compared with the ligation failure after 8-oxodGMP insertion observed in coupled BER assay (Fig. 1c).

**Pol β expression and cellular responses to oxidative stress.** We next tested the idea that oxidative stress leading to oxidized dNTPs in the precursor pool might enable pol β-mediated insertion of oxidized nucleotide into the BER intermediate (Fig. 2a). For this purpose, the effect of pol β expression on cell survival after treatment with the oxidative stress-inducing agent potassium bromate (KBrO₃) was evaluated using isogenic $pol\ \beta^{+/+}$ and $pol\ \beta^{-/-}$ mouse embryonic fibroblast (MEF) cell lines. In addition, we examined cell sensitivity in the absence of activity of the oxidized purine dNTP hydrolase MTH1, either by MTH1 gene deletion or treatment of cells with a MTH1 inhibitor, (S)-crizotinib (Fig. 3). First, we observed more cytotoxicity in $pol\ \beta^{+/+}$ cells than $pol\ \beta^{-/-}$ cells treated with KBrO₃ (Fig. 2b). For both cells treated with KBrO₃ plus (S)-crizotinib, the cytotoxicity was increased (Fig. 3a,b), while (S)-crizotinib alone at the concentration used in the combination studies (6 μM) had minimal effect (Supplementary Fig. 3). MTH1 gene deletion in $pol\ \beta^{+/+}$ ($pol\ \beta^{+/+}MTH1^{-/-}$) cells increased sensitivity (Fig. 3c), while $pol\ \beta^{-/-}MTH1^{-/-}$ cells showed less sensitivity to KBrO₃ (Fig. 3d). Overall, these results obtained where MTH1 was inactivated are consistent with involvement of oxidized nucleotides in the cell sensitivity phenotype associated with KBrO₃ treatment.

Next, we examined strand break formation in $pol\ \beta^{+/+}$ and $pol\ \beta^{-/-}$ cells as a function of cell cycle after KBrO₃ treatment. The results revealed more KBrO₃-induced γH2AX staining (a marker of DNA strand breaks) in both cell lines than in control untreated cells, while less staining was observed in $pol\ \beta^{-/-}$ cells than in $pol\ \beta^{+/+}$ cells (Supplementary Fig. 4). Increased levels of γH2AX were found with increasing KBrO₃ concentrations (15 and 30 mM) and this was consistent with the enhanced cytotoxicity in $pol\ \beta^{+/+}$ versus $pol\ \beta^{-/-}$ cells (Supplementary Fig. 4a). Interestingly, KBrO₃ treatment resulted in γH2AX appearance throughout the cell cycle in both cell lines (Supplementary Fig. 4b) with minimal changes in cell cycle stage (Supplementary Table 3).

**Impact of pol β active site mutation on ligation failure.** We examined the effect of oxidized nucleotide insertion on

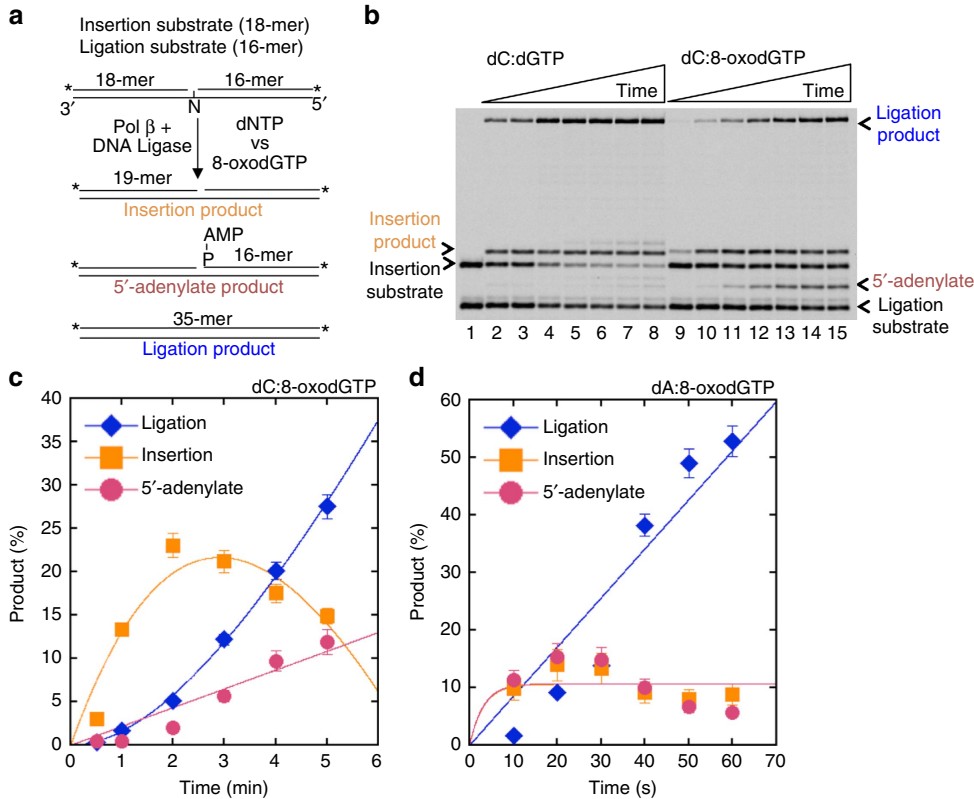

**Figure 1 | Effect of pol β 8-oxodGMP insertion on ligation.** (**a**) Illustrations of DNA substrates and reaction products for insertion, ligation, and 5'-adenylation observed in a coupled BER assay. (**b**) Lane 1 is the minus enzyme control for the substrate $C^{gap}$, lanes 2–8 and 9–15 are the reaction products in the presence of dGTP and 8-oxodGTP, respectively, and correspond to time points of 0.5, 1, 2, 3, 4, 5 and 6 min. Plots show time-dependent changes in the products of 8-oxodGMP insertion, ligation, and 5'-adenylation for the substrates $C^{gap}$ (**c**) and $A^{gap}$ (**d**). Quantification of the products for dC:dGTP is presented in Supplementary Fig. 1a. The data represent mean values with the s.d. from three independent experiments.

ligation for pol β active site mutants R283A and R283K (Fig. 4). The R283 side chain interaction stabilizing the mutagenic 8-oxodGTP(syn) conformation opposite template base A observed in the wild-type enzyme (Fig. 4a) was completely lost in the R283K variant. For the substrate $A^{gap}$, the results with R283K at 10 s incubation showed that production of the 5'-adenylate product was very strong and more persistent compared with wild-type pol β. The effect of the R283A mutation on production of the 5'-adenylate product was a slightly higher than with the lysine mutant (Fig. 4c).

The wild-type R283 side chain stabilizes base pairing in the *anti-anti*-conformations for the template base C, whereas the mutant lysine side chain still interacts with the base pair, but now the base of 8-oxodGTP is closer to the side chain, altering the insertion reaction (Fig. 4b). For the substrate $C^{gap}$, significant differences were observed in the level of 5'-adenylate product between R283K and R283A mutants and compared with wild-type pol β (Fig. 4d).

For both template bases, weaker 8-oxodGMP insertion was observed with the pol β mutants than with wild-type enzyme (Supplementary Fig. 5a,b), while the amount of ligation products showed differences depending on mutant or template base (Supplementary Fig. 5c,d). Overall, these results indicate that ligation failure is triggered by 8-oxodGTP, but without its stable insertion into the gap.

We next investigated the impact of the polymerase-dead pol β D256A mutant on ligation and observed a weak signal for ligation products for both dGMP and 8-oxodGMP insertions (Supplementary Fig. 6a, compare lanes 1–2 with lanes 3–9 and 10–16). This was similar to the weak 5'-adenylation background signal observed in the control reaction with Lig I alone (compare Supplementary Fig. 6b with Supplementary Fig. 2b). These results served as a negative control indicating the D256A enzyme could not stabilize the incoming 8-oxodGTP, and this was not surprising as D256 is required for binding the active site catalytic metal.

**Effect of 3'-8-oxoG and template base on ligation failure.** In control experiments, we verified 5'-adenylation and ligation of the pre-formed 8-oxoG base at the 3'-margin of a nick. The nicked DNA substrates containing template bases A, T, C or G ($A^{nick}$, $T^{nick}$, $C^{nick}$ or $G^{nick}$, Supplementary Table 1) used here mimic the BER intermediates after pol β 8-oxodGMP insertion into the single-nucleotide gapped DNA substrate (Fig. 5a). Template base-dependent differences in 5'-adenylate products were observed (Fig. 5b,c). Ligase was able to ligate the substrates $C^{nick}$ and $A^{nick}$ (Fig. 5d, lanes 3–10 and 12–19), while 5'-adenylate products were considerably more or less, respectively, for the substrates $G^{nick}$ and $T^{nick}$ (Fig. 5e, lanes 3–10 and 12–19).

Pol β 8-oxodGMP insertion coupled with DNA ligation using single-nucleotide gapped DNA substrates containing template bases T or G ($T^{gap}$ or $G^{gap}$, Supplementary Table 1) was also measured. Template base-dependent differences were observed in 5'-adenylate products (Fig. 6a) with relatively more ligation products observed for the substrate $C^{gap}$ (Supplementary Fig. 1b). Quantification of the results indicated time-dependent changes in 5'-adenylate products that were consistent with the coupled insertion and ligation assays described above (compare Fig. 5b,c with Fig. 6a,b).

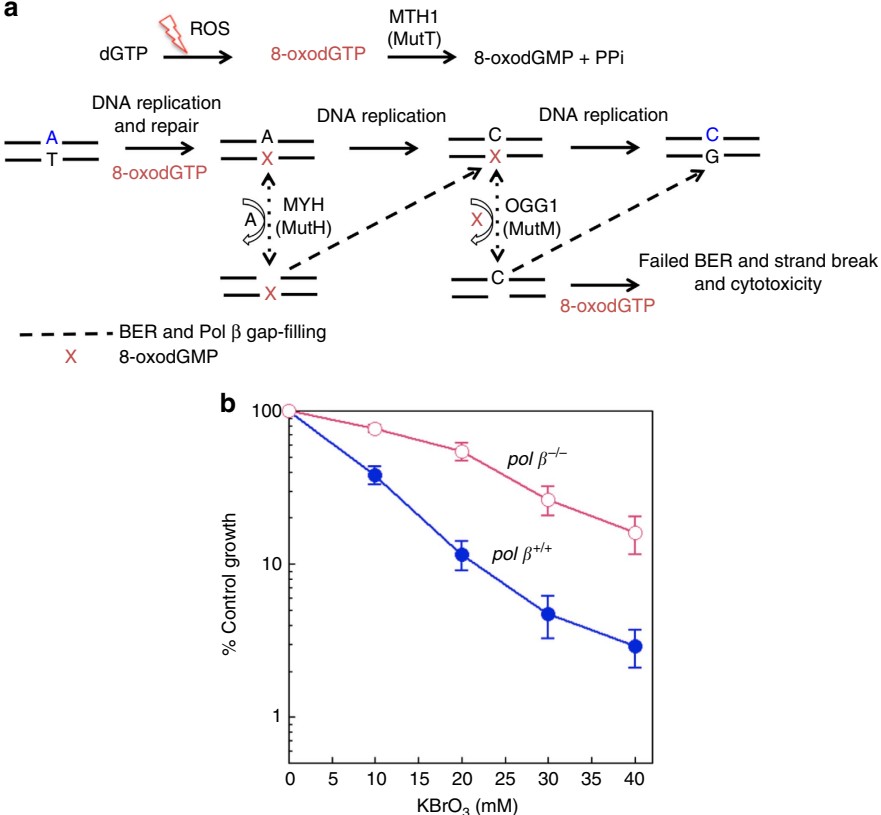

**Figure 2 | Role of pol β in cellular responses to oxidative stress.** (**a**) Illustration of pol β-mediated insertion of 8-oxoGTP upon 8-oxodGTPase deficiency. (**b**) Plots show *pol β*$^{+/+}$ and *pol β*$^{-/-}$ cells treated with a range of concentration of KBrO$_3$ (1 h). The data represent mean values with the s.d. from five independent experiments.

**Comparison of ligation failure for other factors.** In light of the results described above, we tested several other factors that could play a role in impaired BER due to oxidized nucleotide insertion coupled with ligation failure. First, we examined two oxidized adenine bases, that is 8-oxodATP and 2-OH-dATP, in coupled BER reactions with single-nucleotide gapped DNA substrates containing template bases A, T, C or G (A$^{gap}$, T$^{gap}$, C$^{gap}$ or G$^{gap}$, Supplementary Table 1). Similar to 8-oxodGMP, insertions were template base-dependent as well as oxidized adenine base-dependent in terms of levels of products for 5′-adenylation (Fig. 6c,d) and ligation (Supplementary Fig. 1c,d). For the oxidized nucleotides that are matched to a template base, the relative amount of the 5′-adenylate product observed was in the order dC:8-oxodGTP > dT:2-OH-dATP ≫ dT:8-oxodATP. These results are consistent with failed ligation after insertion of these oxidized adenine nucleotides.

Next, we compared ligation failure for Lig I versus the X-ray repair cross-complementing protein 1/DNA Ligase III (XRCC1/Lig III) complex as well as pol β versus pol λ. These comparisons involved 8-oxodGMP insertion opposite template bases C and A (Fig. 7). Similar amounts of 5′-adenylate products were observed in the ligation experiments using the XRCC1/Lig III complex for the substrates C$^{gap}$ (Fig. 7a) and A$^{gap}$ (Fig. 7b) compared with coupled BER reactions including Lig I. For both substrates, slightly more ligation products were observed with the XRCC1/Lig III complex (Supplementary Fig. 7a,b). When the BER back-up DNA polymerase, pol λ, was evaluated, the products for 5′-adenylation and ligation with substrates C$^{gap}$ (Fig. 7c and Supplementary Fig. 7c) and A$^{gap}$ (Fig. 7d and Supplementary Fig. 7d) were found to be similar to pol β.

**Roles of end processing enzymes in repair of impaired BER.** In light of the results described above, we examined the fate of nicked BER intermediates containing lesions at the 3′- and 5′-margins of the nick (Substrates 1 and 2, Supplementary Table 1). Results were obtained with four potential end-processing enzymes as follows: AP endonuclease 1 (APE1), tyrosyl phosphodiesterase 1 (Tdp1), aprataxin (APTX) and flap endonuclease 1 (FEN1). APE1 and Tdp1 were able to remove 3′-margin 8-oxoG along with additional excision products (Supplementary Fig. 8a); APTX was capable of editing the 5′-adenylated BER intermediates by removing the 5′-AMP moiety, and FEN1 excised the 5′-adenylate along with a nucleotide flap and significant amounts of other excision products were also produced (Supplementary Fig. 8b). In addition to these BER end-processing enzymes, we evaluated the DNA glycosylases 8-oxoguanine DNA glycosylase (OGG1), Nei Endonuclease VIII-Like 1 (NEIL1), and Endonuclease VIII (NTH1) for their ability to remove the nick margin 3′-8-oxoG. These enzymes were found to be not active against the nicked substrate with 3′-margin 8-oxoG and 5′-margin AMP lesions (Supplementary Fig. 8c).

**Processing of impaired BER by cell extracts.** Finally, we investigated processing of 3′-8-oxoG and 5′-AMP-containing BER intermediate in whole-cell extracts from wild type and APTX-deficient DT40 cell lines (Supplementary Fig. 9a). In the reference reactions using purified enzymes, products of APTX and FEN1 activities, as described above, were evaluated for comparison (Supplementary Fig. 9b, lanes 2 and 3). Using cell extracts from wild-type cell lines, products similar to those in the

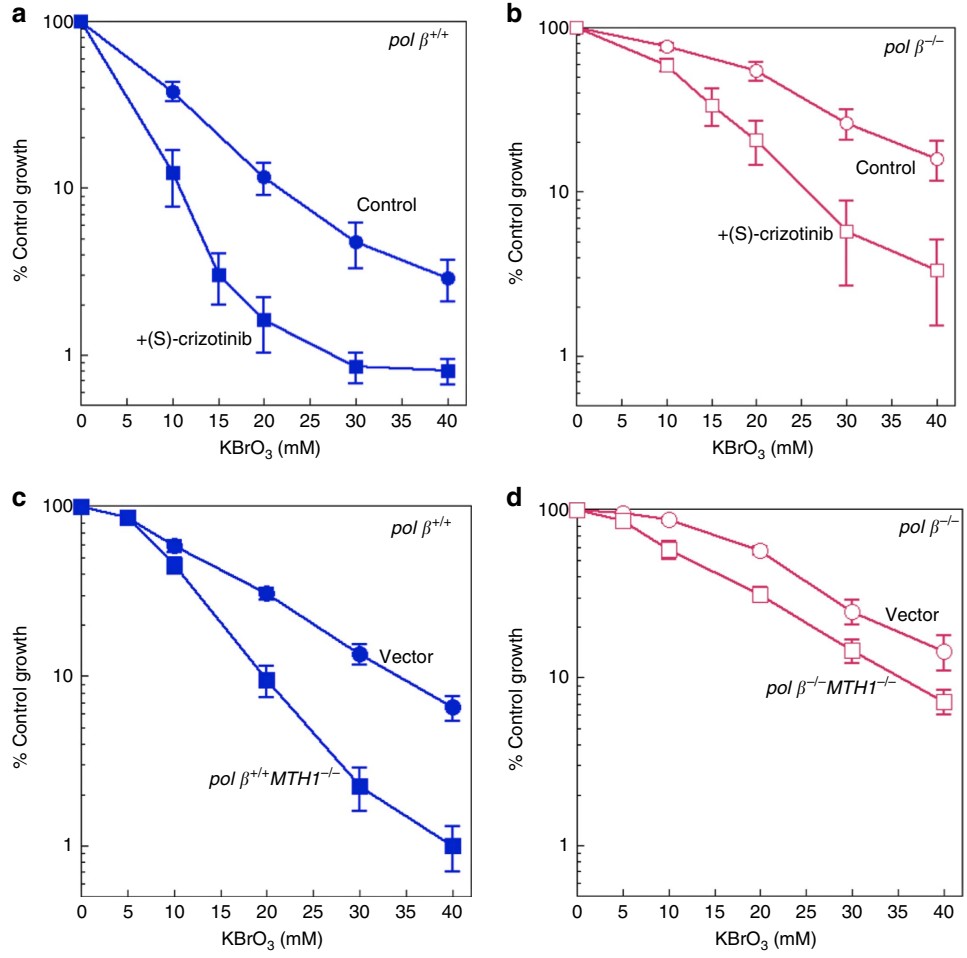

**Figure 3 | Sensitivity of MEFs to oxidative stress. (a,b)** Plots show *pol β*$^{+/+}$ and *pol β*$^{-/-}$ cells treated with a range of concentration of KBrO$_3$ (control) and KBrO$_3$ plus (S)-crizotinib (6 μM) continuously. The data represent mean values from three independent experiments. The effect of (S)-crizotinib alone (0–10 μM) in *pol β*$^{+/+}$ and *pol β*$^{-/-}$ cells is presented in Supplementary Fig. 3. Sensitivity of *pol β*$^{+/+}$ vector and *pol β*$^{+/+}$ *MTH1*$^{-/-}$ (**c**), and *pol β*$^{-/-}$ vector and *pol β*$^{-/-}$ *MTH1*$^{-/-}$ (**d**) cells to a 1 h exposure to KBrO$_3$. Cells transfected with empty lentivirus vector, Lenti-eSpCas9, were used as controls for analysis. The data represent mean values with the s.d. from three or four independent experiments.

reference reactions were observed (Supplementary Fig. 9b, lanes 4–9). Using extracts from APTX-deficient cells, the results indicated that FEN1 removal of 5′-AMP plus one nucleotide was observed along with weak bands corresponding to 5′-AMP plus two or more nucleotides (Supplementary Fig. 9b, lanes 10–15). These results confirm that APTX and FEN1 provide alternative reactions for removing the blocking 5′-AMP group in presence of 3′-8-oxoG lesion and FEN1 activity, that is, long-patch BER sub-pathway, in the cell extracts was strong enough to complement the deficiency in APTX activity.

## Discussion

Oxidation of NTPs and dNTPs in the nucleotide pool can be a source of RNA and DNA damage when DNA polymerases incorporate these oxidized nucleotides into RNA and DNA[8,10,24,25]. Structural and biochemical results have supported the hypothesis that many DNA polymerases discriminate between alternate nucleotide substrates through an induced fit mechanism where binding of the correct nucleotide leads to conformational adjustments that align active site catalytic groups to optimize chemistry[26,27]. For example, X-ray crystallography studies revealed that pol β can close around the nascent base pair and insert 8-oxodGMP(*anti*) opposite template base C(*anti*)[28–30]. Moreover, time-lapse snapshots of the enzyme showed that after this insertion event, the Watson–Crick base pair is lost, and the newly inserted 8-oxodGMP base adopts a frayed position stacked on top of the template base[16]. This arrangement was reminiscent of a newly inserted mismatched nucleotide in the active site[16]. In the present study, we examined the effects of pol β-mediated oxidized nucleotide insertion on downstream steps in the BER pathway. Our working model for BER suggests that after pol β fills the gap, the resulting nicked BER intermediate is passed to a DNA ligase for completion of repair[23]. Thus, the effect of oxidized nucleotide insertion on the ligation step is an important topic of investigation. This could be a critical issue in the case of pol β insertion since it lacks 3′-exonuclease editing activity.

In this study, we showed that pol β insertion of oxidized deoxynucleotide leads to ligation failure along with formation of the 5′-adenylate product. We found differences in failed ligation based on the identity of template base and type of oxidized purine nucleotide. This could either reflect cross-talk between pol β and DNA ligase within the BER intermediate to which both enzymes bind or could be due to structural differences in interaction between two enzymes as has been suggested by other studies[31,32]. Moreover, mutations in the polymerase active site could affect this interaction, as

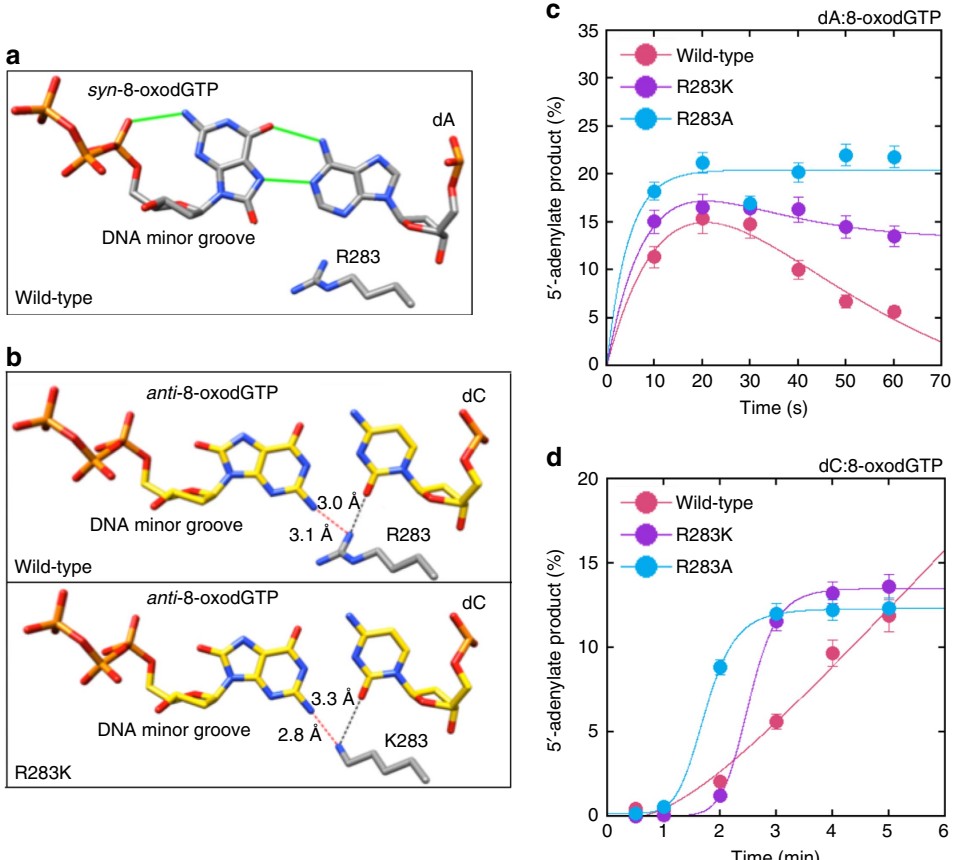

**Figure 4 | Comparison of ligation failure for pol β active site mutants R283A and R283K.** Structural images show pre-catalytic ternary substrate complex with 8-oxodGTP as the incoming nucleotide opposite template bases A (**a**) and C (**b**) in a single-nucleotide gap. Plots show differences in the products of 5′-adenylation between wild type and pol β mutants for the substrates A[gap] (**c**) and C[gap] (**d**). Time-dependent changes in insertion and ligation products are presented in Supplementary Fig. 5. The data represent mean values with the s.d. from three independent experiments.

shown by formation of stable and strong 5′-adenylation products with the pol β R283K and R283A mutants. Overall, the in vitro results described here indicated that the BER intermediate with the 3′-margin 8-oxoG lesion could represent an oxidative stress-induced stalled repair intermediate capable of triggering cytotoxicity, and the resulting BER intermediate with blocks at both 3′- and 5′-ends may be a source of stalled repair intermediates[33].

In the cell-based studies decribed, KBrO₃ was selected as oxidizing agent since it is known to cause DNA damage through oxidative stress and formation of reactive metabolites that preferentially oxidize guanine residues[34]. Our in vivo findings indicated KBrO₃-induced hypersensitivity in $pol \, \beta^{+/+}$ cells compared with $pol \, \beta^{-/-}$ cells. In our previous work[35], we reported that $pol \, \beta^{+/+}$ cells are more resistant than $pol \, \beta^{-/-}$ cells to an another oxidizing-agent hydrogen peroxide ($H_2O_2$). We believe the difference is likely due to the nature of the oxidizing agent-mediated DNA damage in cells. $H_2O_2$ has been widely utilized as a representative ROS for studies of oxidative stress in mammalian cells and is considered to induce single-strand breaks and various forms of base damage[36–38]. Thus, we considered $H_2O_2$ as less likely than KBrO₃ to specifically oxidize guanine in the nucleotide pool. This effect likely involves oxidized nucleotide accumulation in the nucleotide pool and is consistent with an adverse effect of pol β mediated oxidized nucleotide insertion and attendant accumulation of stalled BER intermediates. This idea remains to be investigated in more detail. We further examined KBrO₃-induced cytotocity in

both $pol \, \beta^{+/+}$ and $pol \, \beta^{-/-}$ cells with MTH1 gene deletion. MTH1 is an oxidized purine dNTP hydrolase. The results revealed higher cytotoxicity in $pol \, \beta^{+/+} \, MTH1^{-/-}$ cells than in $pol \, \beta^{-/-} \, MTH1^{-/-}$ cells. These results are consistent with the KBrO₃-induced cell killing effect observed in $pol \, \beta^{+/+}$ cells co-treated with the MTH1 inhibitor (S)-crizotinib[39]. One explanation for the cell killing phenotype observed in the absence of pol β could involve backup BER DNA polymerases able to incorporate oxidized nucleotides. We showed similar ligation failure coupled with pol λ 8-oxodGMP insertion. In addition, the long-patch BER sub-pathway may be involved in repair of oxidative DNA damage, as we determined a potential role of FEN1 to function on the 5′-margin adenylated intermediate and its complementation activity in the extracts from APTX-deficient cells[11,15,18,19].

In addition to the in vivo findings reported here, the importance of the pol β-dependent BER pathway in repairing oxidative damage has been reported. Increased levels of pol β and BER activity in mice were induced by the oxidative stress agent 2-NP (ref. 40). Moreover, preferential repair of 8-oxoG-containing DNA by short-patch BER in $pol \, \beta^{+/+}$ versus $pol \, \beta^{-/-}$ cell extracts from HeLa and MEF cells has been observed[41–44]. Overall, these findings suggest that pol β can mediate BER of oxidative stress-induced DNA damage in cells. The reduced incorporation of radiolabeled 8-oxodGTP into a gapped plasmid in cell extract from pol β null cells and the increased incorporation by addition of purified pol β are consistent with oxidized nucleotide BER in intact cells[45].

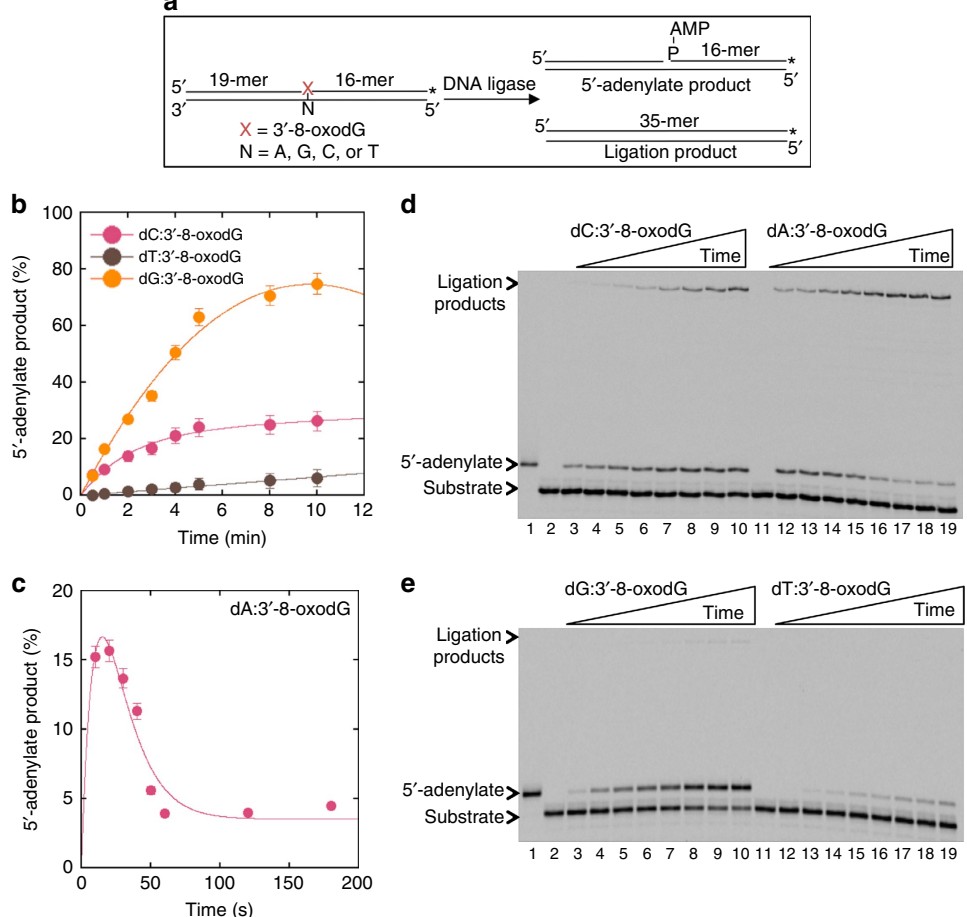

**Figure 5 | Effect of 3′-8-oxoG on ligation. (a)** Illustrations of nicked DNA substrate and reaction products for ligation and 5′-adenylation. Plots show time-dependent changes in the products of 5′-adenylation for the substrates C$^{nick}$, G$^{nick}$, T$^{nick}$ **(b)** and A$^{nick}$ **(c)**. The data represent mean values with the s.d. from three independent experiments. **(d)** Lane 1 is the 5′-adenylated oligonucleotide used as a marker, lanes 2 and 11 are the minus enzyme controls for the substrates C$^{nick}$ and A$^{nick}$, respectively. Lanes 3–10 and 12–19 are the reaction products, and correspond to time points of 0.5, 1, 2, 3, 4, 5, 8, 10 min, and 10, 20, 30, 40, 50, 60, 120, 180 s, respectively. **(e)** Lane 1 is the 5′-adenylated oligonucleotide used as a marker, lanes 2 and 11 are the minus enzyme controls for the substrates G$^{nick}$ and T$^{nick}$, respectively. Lanes 3–10 and 12–19 are the reaction products, and correspond to time points of 0.5, 1, 2, 3, 4, 5, 8 and 10 min.

Although OGG1 is the main DNA glycosylase for excising 8-oxoG from DNA in mammalian cells[18], the purified enzyme had no activity against a substrate with 3′-margin 8-oxoG, that is, a substrate that mimics the nicked BER intermediate after pol β 8-oxodGMP insertion and ligation failure. We note, however, that limited activity of DNA glycosylases OGG1, NEIL1 and NTH1 on oxidative base lesions at the 3′-end of single-stranded breaks was reported[46–48]. The role of APE1 in excision of 3′-8-oxoG lesions in human cell extracts has been described[48], and we showed here that both APE1 and Tdp1 have activity against 3′-8-oxoG at a nick.

In summary, our study demonstrates the role of pol β-oxidized nucleotide insertion in the impairment of BER due to the ligation failure after the insertion, as revealed by formation of toxic BER intermediates with blocked 3′-8-oxoG and 5′-AMP-including ends. We also note the potential for incorporation of 8-oxodGMP during DNA replication leading to genomic 8-oxodG. The repair of these lesions may result in re-insertion of 8-oxodG and production of cytotoxic strand breaks. Further studies, that is, oxidized dNTP pool measurements from the cells used in this study, will be necessary at the biological level to enhance our understanding for role of pol β in oxidative stress-induced damage response in cells.

## Methods

**Coupled BER assay with pol β and ligase.** The DNA substrates with template bases A, T, C or G (A$^{gap}$, T$^{gap}$, C$^{gap}$ or G$^{gap}$, Supplementary Table 1) were prepared as follows[49,50]: The 5′-end FAM-labelled upstream oligonucleotide (19-mer) was annealed with the 3′-end FAM-labelled downstream oligonucleotide (17-mer) in the presence of the template oligonucleotide (37-mer), to generate a single-nucleotide gapped DNA substrate. Coupled BER assays including both pol β and Lig I or pol β and XRCC1/Lig III complex were conducted under steady-state conditions where the gapped DNA substrate was in excess (250 nM) over the enzyme mixture (100 nM). The reaction was initiated by addition of both enzymes that had been preincubated at 37 °C for 3 min. The reaction mixture in a final volume of 10 μl contained 50 mM Tris(Hcl), pH 7.5, 100 mM KCl, 10 mM MgCl$_2$, 1 mM ATP, 100 μg ml$^{-1}$ BSA, 10% glycerol, 1 mM DTT and 200 μM of an oxidized base nucleotide (8-oxodGTP (Jena Bioscience), 2-OH-dATP (Jena Bioscience) or 8-oxodATP (TriLink Biotechnologies)) or a normal base nucleotide (dGTP, dTTP, dATP, or dTTP (NEB)). The coupled BER assays with pol β mutants R283K, R283A or D256A plus Lig I were performed similarly. The incubation was at 37 °C for indicated time points. The reactions were then quenched with 0.3 mM EDTA and mixed with an equal volume of gel loading buffer (95% formamide, 20 mM EDTA, 0.02% bromphenol blue, and 0.02% xylene cyanol). After incubation at 95 °C for 3 min, the reaction products were separated by electrophoresis in a 15% polyacrylamide gel containing 8 M urea in 89 mM Tris-HCl, 89 mM boric acid and 2 mM EDTA, pH 8.8. The gels were scanned on a Typhoon PhosphorImager, and the data were analysed using ImageQuant software.

**Ligation assays.** Synthetic oligodeoxyribonucleotide with 3′-8-oxoG was from Integrated DNA Technologies, Inc. The 5′-end of the oligodeoxyribonucleotide was

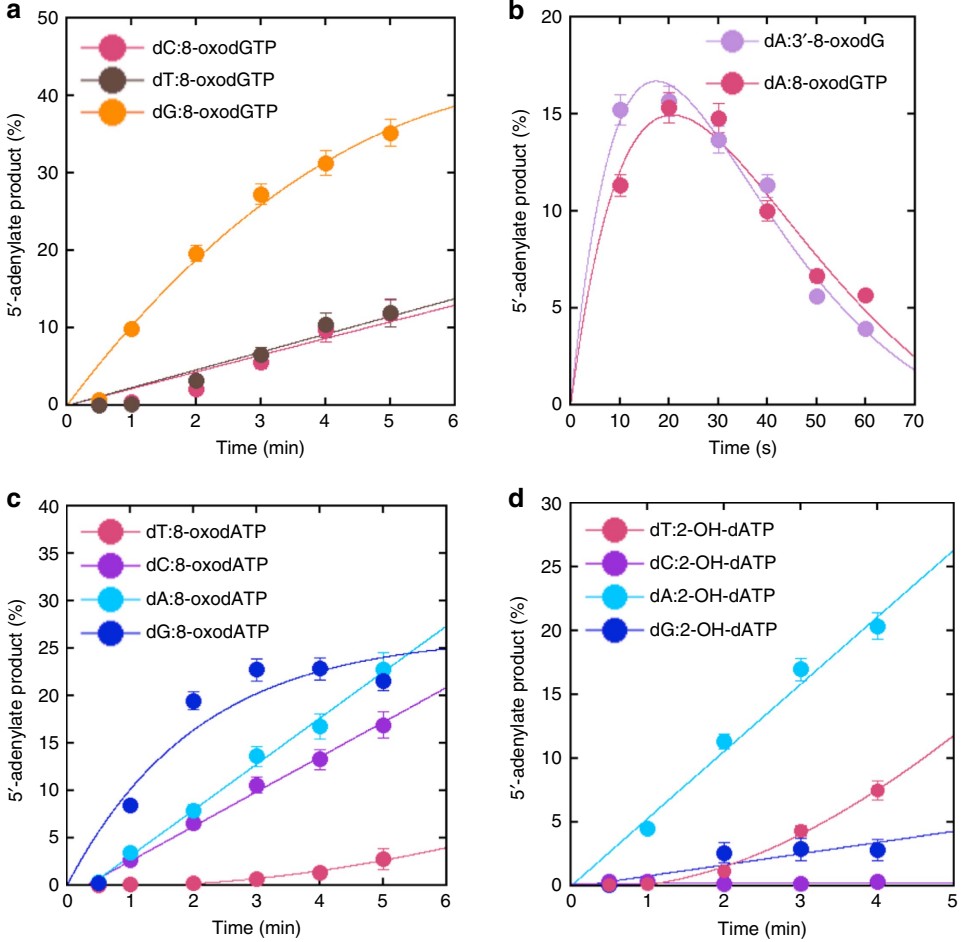

**Figure 6 | Template base and oxidized nucleotide-dependent comparison of ligation failure.** Plots show template base-dependent changes in the products of 5′-adenylation for the substrates $C^{gap}$, $G^{gap}$, and $T^{gap}$ (**a**), $A^{gap}$ and $A^{nick}$ (**b**), oxidized nucleotide and template base-dependent changes in the products of 5′-adenylation for 8-oxodATP (**c**) and 2-OH-dATP (**d**) for the substrates $A^{gap}$, $C^{gap}$, $G^{gap}$ and $T^{gap}$. Time-dependent changes in ligation products are presented in Supplementary Fig. 1b–d. The data represent mean values with the s.d. from three independent experiments.

adenylated using the 5′-DNA adenylation kit[49]. Briefly, the adenylation reaction was performed in a reaction mixture (20 μl) containing 100 pmol oligonucleotide, 50 mM sodium acetate, pH 6.0, 10 mM MgCl₂, 5 mM DTT, 0.1 mM EDTA and 0.1 mM ATP, and the reaction was initiated by adding 100 pmol *Mth* RNA ligase and incubated at 65 °C for 1 h. After heat inactivation of the enzyme at 85 °C for 5 min, the adenylated oligonucleotides were separated and then purified from other DNA species by electrophoresis in a 18% polyacrylamide gel containing 8 M urea in 89 mM Tris-HCl, 89 mM boric acid and 2 mM EDTA (pH 8.8). The DNA substrates with template bases A, T, C or G ($A^{nick}$, $T^{nick}$, $C^{nick}$ or $G^{nick}$, Supplementary Table 1) were prepared as follows[49,50]: the upstream oligonucleotide with 3′-8oxoG (18-mer) was annealed with the 3′-end FAM-labelled downstream oligonucleotide (16-mer) in the presence of the template oligonucleotide (34-mer) to generate the nicked DNA substrates. The ligation assays were performed in a reaction mixture (10 μl) containing 50 mM Tris(HCl), pH 7.5, 100 mM KCl, 10 mM MgCl₂, 1 mM ATP, 100 μg ml⁻¹ BSA, 10% glycerol, and 250 nM DNA substrate. The reaction was initiated by addition of Lig I (100 nM), incubated at 37 °C for indicated time points, and stopped with an equal volume of 95% formamide dye. The reaction products were separated and quantified as described above.

**Repair assays with end processing enzymes.** The nicked DNA substrates used for activity assays of 5′- and 3′-end processing enzymes (Substrates 1 and 2, Supplementary Table 1) were prepared as follows[49,50]: for DNA substrate 1, the 5′-end FAM-labelled upstream oligonucleotide with 3′-8oxoG (19-mer) was annealed with the downstream oligonucleotide with 5′-AMP (16-mer) in the presence of the template oligonucleotide (35-mer). For DNA substrate 2, the upstream oligonucleotide with 3′-8oxoG (19-mer) was annealed with the 3′-end FAM-labelled downstream oligonucleotide with 5′-AMP (16-mer) in the presence of the template oligonucleotide (35-mer). The components of the reaction mixtures are as follows: for NTH1 and NEIL1 activities[47]: 50 mM

HEPES(KOH), pH 7.8, 50 mM KCl, 10 mM MgCl₂, 0.5 mM EDTA, 8.5% glycerol, 1.5 mM DTT and 100 μg ml⁻¹ BSA; for APE1 activity[48]: 25 mM HEPES(KOH), pH 7.9, 100 mM KCl, 12 mM MgCl₂, 0.1 mM EDTA, 17% glycerol and 2 mM DTT; for OGG1 activity[51]: 50 mM HEPES, pH 7.5, 20 mM KCl, 0.5 mM EDTA and 0.1% BSA; for Tdp1 activity[52]: 50 mM Tris(HCl), pH 7.0, 80 mM KCl, 5 mM MgCl₂, 2 mM EDTA, 1 mM DTT and 40 μg ml⁻¹ BSA; for APTX activity[49,50]: 50 mM HEPES, pH 7.5, 20 mM KCl, 0.5 mM EDTA and 2 mM DTT; and for FEN1 activity[49,50]: 50 mM HEPES, pH 7.5, 50 mM KCl, 10 mM MgCl₂ and 0.5 mM EDTA. For both end processing enzymes, the reaction mixture of final volume (10 μl) contained 200 nM DNA, the indicated amount of the enzyme, and incubated at 37 °C for 30 min (3′-end processing enzymes) or 15 min (5′-end processing enzymes). The reaction products were separated and quantified as described above.

**Enzymatic assays in cell extracts.** The activity for 5′-end processing enzymes was measured using the cell extracts from wild type and APTX-deficient DT40 cells[50]. The repair assays for reference reactions including purified proteins FEN1 and APTX were performed as described above. For reaction mixtures including cell extracts, the reaction was initiated by addition of cell extract (50 μg) in a reaction mixture (10 μl) containing 250 nM DNA substrate (Substrate 2, Supplementary Table 1), 50 mM HEPES, pH 7.5, 50 mM KCl, 10 mM MgCl₂ and 0.5 mM EDTA. The reaction was stopped at the indicated time points with an equal volume of 95% formamide dye. The reaction products were separated and quantified as described above.

**Lentiviral vector construction for MTH1 knockout.** To construct a lentivirus vector containing the enhanced specificity SpCas9 (eSpCas9)[53], the expression cassette of eSpCas9 was amplified from plasmid eSpCas9 (1.1; Addgene plasmid ID: 71814) and was then double-digested with endonucleases XbaI (NEB) and

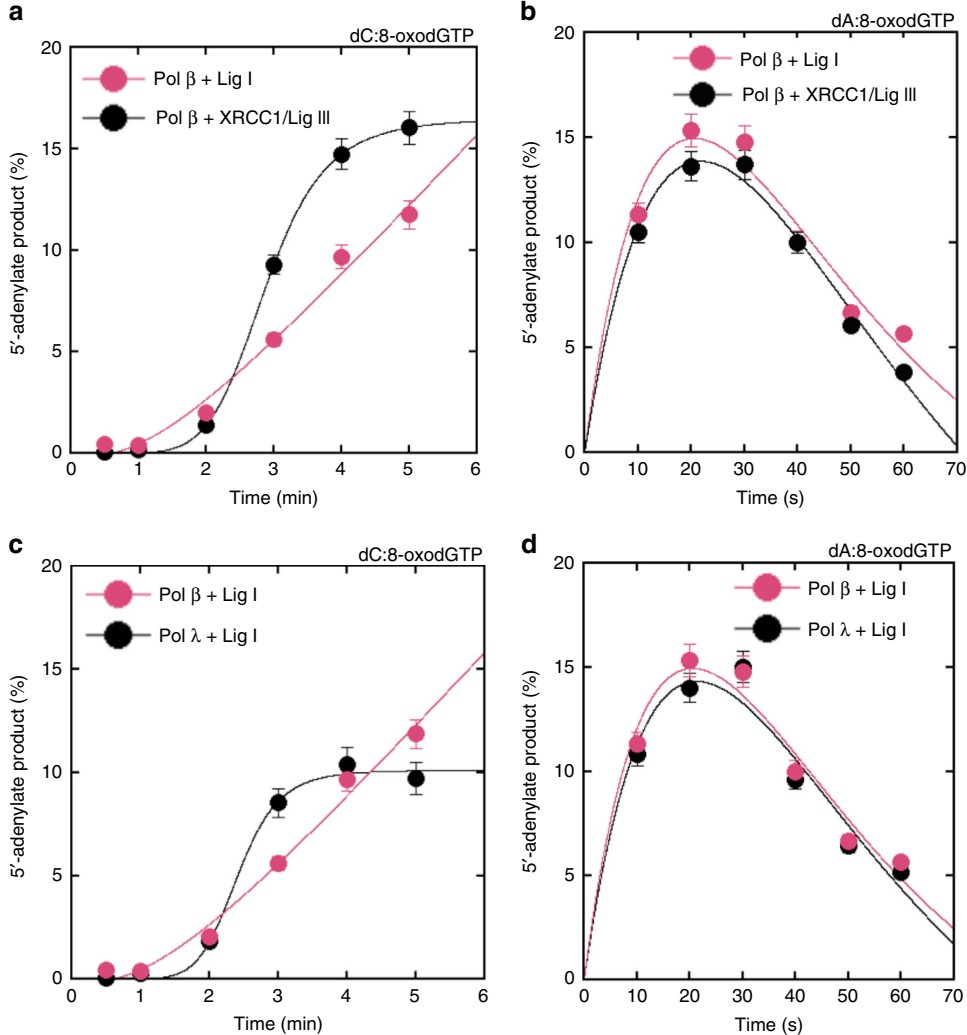

**Figure 7 | Effect of XRCC1/Lig III and pol λ on ligation failure.** Plots show time-dependent changes in the products of 5′-adenylation for Lig I versus XRCC1/Lig III complex for the substrates $C^{gap}$ (**a**) and $A^{gap}$ (**b**) and for pol β versus pol λ for the substrates $C^{gap}$ (**c**) and $A^{gap}$ (**d**). Time-dependent changes in ligation products are presented in Supplementary Fig. 7. The data represent mean values with the s.d. from three independent experiments.

BamHI (NEB). After purification, the product was cloned into the XbaI/BamHI site within the lentivirus vector LentiCRISPR v2 (Addgene plasmid ID: 52961). The final expression vector was termed Lenti-eSpCas9. CRISPR Design Tool (http://tools.genome-engineering.org) was used to predict the suitable MTH1-targeting sgRNA guide sequences. Oligonucleotides or PCR primers for vector construction and PCR amplification were from Integrated DNA Technologies (Supplementary Table 2).

The sgRNA guide sequences (T1-T4) were cloned into the expression vector as follows (Supplementary Fig. 10a). The Lenti-eSpCas9 plasmid was cut and dephosporylated with FastDigest BsmBI and FastAP (Fermentas) at 37 °C for 2 h. After purification with the QIAEX Gel Extraction kit (Qiagen), 50 ng of vector was ligated to 50 ng of pre-annealed sgRNA oligonucleotide using a DNA ligation kit (Takara Bio) at 16 °C for 30 min. Then, the mixture was transformed into the *E. coli* competent strain Stbl3 (Invitrogen) according to the protocol supplied with the cells. To verify that transformed plasmids had the correct inserted sequences, each colony was sequenced from the U6 promoter using primer hU6-F (Supplementary Table 2).

**Lentiviral production and cell colony selection.** The lentivirus was prepared and packaged as follows[54]: The Lenti-eSpCas9 vectors were co-transfected with packaging plasmids pMD2.G (Addgene plasmid ID: 12259) and psPAX2 (Addgene plasmid ID: 12260) as previously reported, MEF cells were then infected with Lenti-eSpCas9 virus at MOI = 1 and selected for 5 days with 2.5 μg ml$^{-1}$ puromycin (Life Technologies) to eliminate untransformed cells.

A portion of the puromycin-resistant cells was used for DNA extraction and SURVEYOR assay[54,55] to find the suitable target guide sequences with high genome-editing efficiency. Another group of puromycin-resistant cells was dissociated by dilution, and single cells were seeded into separate wells in 96 well plates for continuous selection in culture medium supplement with 2.5 μg ml$^{-1}$ puromycin until the candidate clones were grown up for further mutation detection. Cells transfected with empty vector Lenti-eSpCas9 were also selected as vector control cells expressing normal level of MTH1 protein.

**Colony verification.** The SURVEYOR assay was carried out with a SURVEYOR mutation detection kit S100 (Integrated DNA Technologies) to detect mutated cells caused by the CRISPR-Cas system as follows[54,55]: the QuickExtract DNA extraction solution (Epicentre) was used to rapidly extract DNAs. This was used for the PCR amplification of exon1 of the *MTH1* gene with the primers (Supplementary Table 2). The PCR product was then mixed with the amplicon of wild type to form heteroduplex by gradient annealing, and the heteroduplex was digested by Surveyor nuclease S in SURVEYOR mutation detection kit at 42 °C for 30 min. The SURVEYOR nuclease digestion products were visualized on a 2% (wt/vol) agarose gel. Positive clones in the SURVEYOR assay were kept, and cells were grown up to obtain enough material for Sanger sequencing and western blot confirmation. For Sanger sequencing (Supplementary Fig. 11), PCR products in the SURVEYOR assay were purified with QIAquick PCR purification kit (Qiagen) and used for sequencing by primer mMTH1-seq (Supplementary Table 2). Western blot analysis was performed as follows: 60 μg protein was loaded and separated with a 15% Criterion Tris-HCl Gel (Bio-Rad), the samples were then transferred to nitrocellulose membrane (Life Technologies) and immunoblotted with rabbit polyclonal anti-MTH1 antibody (Abcam, ab197855) or anti-pol β antibody in 1:2,000 dilution (Supplementary Fig. 10b).

**Cell culture.** Clones of isogenic $pol\ \beta^{+/+}$ and $pol\ \beta^{-/-}$ MEF cell lines (36.3 and 38.4, respectively)[56] were used for this study. Cells were grown at 34 °C in a 10% $CO_2$ incubator in DMEM supplemented with GlutaMAX-1 (Gibco), 10% fetal bovine serum (FBS, HyClone), hygromycin (80 μg ml$^{-1}$), and were routinely tested and found to be free of mycoplasma contamination. Cytotoxicity was determined by growth inhibition assays as follows[57]: briefly, cells were seeded in six-well dishes at a density of 40,000 cells per well, and treated for 1 h with a range of concentrations of $KBrO_3$ in the absence or presence of 6 μM (S)-crizotinib (Sigma-Aldrich). After washing, cells were further incubated as appropriate with (S)-crizotinib until counting of nuclei collected from the plates (triplicate wells for each drug concentration) following cell lysis using hypotonic solution then detergent[58]. Results were expressed as the number of cells in $KBrO_3$-treated wells relative to untreated cells (% control growth). Clones of both $pol\ \beta^{+/+}$ and $pol\ \beta^{-/-}$ vector-expressing and MTH1 knockout were similarly plated and treated at 37 °C for 1 h with a range of concentrations of $KBrO_3$. Sensitivity was determined by growth inhibition assays as described above.

**γH2AX phosphorylation assay.** Cells were seeded in 100 mm dishes at a density of $1 \times 10^6$. The following day, cells were treated with $KBrO_3$ (15 and 30 mM) for 1 h, washed once, and incubated with medium for 3 h. Cells were then collected by trypsinization, centrifuged, and washed with PBS. The cell pellet was prepared for flow cytometry using a γH2AX phosphorylation assay kit (EMD Millipore) with the addition of propidium iodide to counterstain DNA and reveal cell cycle phase. Samples were read on an LSR II (BD Biosciences) and analysed using FACSDiva software (BD Biosciences).

**Data availability.** The data that support the findings of this study are available upon reasonable request.

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

## Acknowledgements

We thank Rajendra Prasad, David Shock, Akira Sassa, Nadezhda Dyrkheeva, Alan Tomkinson (Univ. of New Mexico) and Stephen Lloyd (OHSU) for gifts of purified proteins used in this study. We thank Carl Bortner for his advice with Flow Cytometry and Natalie Gassman for helpful discussion. This work was supported by the Intramural Research Program of the NIH, National Institute of Environmental Health Sciences (Project numbers Z01 ES050158 and Z01 ES050159).

## Author contributions

M.Ç. performed the *in vitro* biochemistry experiments. J.K.H. and D.F.S. conducted the *in vivo* cell studies. D.-P.D. engineered and characterized the *MTH1* gene deletion. M.Ç. and S.H.W wrote the manuscript.

## Additional information

**Competing financial interests:** The authors declare no competing financial interests.

**How to cite this article:** Çağlayan, M. *et al.* Oxidized nucleotide insertion by pol β confounds ligation during base excision repair. *Nat. Commun.* **8,** 14045 doi: 10.1038/ncomms14045 (2017).

