## [Peer Review File · Nature Communications]

Reviewers' Comments:

Reviewer #1 (Remarks to the Author)

Manuscript: "Oxidized nucleotide insertion by pol β confounds ligation during base excision repair."

Summary: This manuscript examines the threat of oxidized nucleotide incorporation during BER to abortive ligation. The majority of the work is biochemical with a small contribution using cells to demonstrate that oxidized nucleotides do in fact, result in greater cytotoxicity and increased γ H2AX, dependent on pol β . The work stresses the damaging effects of incomplete BER, and how continued oxidative exposure may increase the likelihood of more serious problems such as DNA breaks. The authors present a model in which the insertion of oxidized dNTPs into DNA by Pol β during BER results in the generation of 5'-adenylated DNA, DNA ligation failure, and the cytotoxic DNA breaks. The manuscript is well written and the in vitro experiments are nicely carried out and clearly presented. However, additional in vitro and cell experiments are needed to substantiate the results and the main conclusions stated in the manuscript.

Figure 2 clearly shows that wild-type cells are more sensitive to potassium bromate-mediated oxidative stress than pol β null cells, and that the effect is compounded by the inhibition of MTH1. The authors conclude that increased oxidative nucleotide pools contribute to pol β -mediated incorporation of oxidized nucleotides, resulting in cellular sensitivity. This is a logical conclusion, but in Figure 7, the authors demonstrate that pol λ , the backup enzyme for pol β , behaves similarly in regards to insertion of oxidized nucleotides as well as 5'-adenylate formation. It seems that pol β null cells, not acutely knocked down for pol β , but chronically depleted, should also experience abortive ligation due to pol λ activity. If the cells in question do not express λ in sufficient amounts, this may be a more trivial issue to address.

Also regarding Figure 2, a paper from this same lab shows that pol β null, pol λ null, and particularly the double mutant are markedly sensitive to hydrogen peroxide. The effects of KBrO₃ are clearly different, but the text should explain the difference in those agents.

Figure 2C.

An experiment should be included with and without (S)-crizotinib and no KBrO₃ treatment to get detect the steady-state concentration of oxidized dNTPs in the nucleotide pool in these cells and its effect on cell survival.

The Experiment in Fig. 2c should also be carried out following Mth1knockdown or knockout to demonstrate the specificity of (S)-crizotinib and that Mth1 inhibition is responsible for the enhanced cytotoxicity.

Figures 3 and 4 appear to present the same data in multiple graphs. For example, Figure 4d is the combination of R283K data from 3d and R283A data from 4c. The same is true with the 4b, 3b, and 4a. This should be presented in a single graph with WT, R283K and R283A. For clarity, the graph could be a bit larger than presented now and still take up less space than 3 graphs containing duplicated information.

The association between supplemental figure 3 and figure 5 could be made more clear if the authors included "quantitated in figure 5a" or similar language. Alternatively, supplementary figure 3 could be added to the main part of the paper for more clarity.

Results of OGG1, NEIL1, and NTH1 should at least be included in supplement as they further demonstrate the risk of 8-oxoG incorporation in a BER intermediate as the nick can't be ligated or

repaired.

Supplementary Figure 2.

Mth1 inhibition by (S)-crizotinib, Mth1 knockdown, and no KBrO₃ treatment, could be included to demonstrate the biological significance of oxidized dNTPs insertion by Pol β and the generation of DNA strand breaks detected by gamma-H2AX staining.

Supplementary Figure 4.

The purpose of these experiments is not obvious. However, an experiment to test the consequence of 8oxodGMP incorporation by Pol β in the presence of one or several of these enzymes in the reaction either individually or in combination could be considered.

- Lane 159: "...Substrates 3 and 4..." do you mean substrates 1 and 2?

Finally, the discussion immediately drops off without any statement of summary or significance.

Reviewer #2 (Remarks to the Author)

The manuscript "Oxidized nucleotide insertion by pol beta confounds ligation during base excision" by Caglayan et al describes how insertion of oxidized purine nucleotides by polymerase beta can compromise ligation in BER. This study is a follow up of the recent Nature report by the same group: "Uncovering the polymerase-induced cytotoxicity of an oxidized nucleotide", now showing the consequences of polymerase beta executed incorporation of oxidized nucleotides in in vitro assays and some in vivo (cellular) data.

The manuscript presents important findings. The authors uncover the potential challenges and pitfalls of oxidative damage repair; a process that affects aging, among others, and highlights the role of BER and pol beta in this context. The data support most of the conclusions made by the authors. The data, methodology and technical execution is of high quality. Proper controls have been applied and shown. Uncertainties have been largely addressed. Even though informative and exposing potentially important elements in repair, it is not clear however, how the mechanisms revealed in the in vitro assays (when only pol beta and ligases are present) relate to cellular repair. This reviewer supports publication of this study, however, strongly suggests the following to elevate their findings with regards to biological significance.

- Even though 5-AMP formation is evident and is clearly related to oxidized nucleotide incorporation, ultimately, ligation occurs to a large part. (Please add dC:dGTP quantification to Figure 1 for comparison!) Also, with few exceptions, 5'adenylate products amount to about 10-20% only. How do the authors envision biological significance in the context of cellular repair that also incorporates end processing enzymes and other BER elements, notably APTX? The supplementary data show a role for APTX in 5'AMP removal – it is not really clear how these in vitro assays were exactly performed, please clarify. The same authors reported compensatory mechanisms before in 2014. In detail analysis when testing oxidized nucleotide incorporation with respect to APTX's role could be a valuable addition.

- In order to provide a role of their findings for cellular repair, the authors treated cells that express wildtype pol beta or were deficient in pol beta with KBrO₃. This reviewer strongly advises to substantiate these data, so to minimize potential artefactual conclusions. Cellular models should include pol beta complemented cells. This also applies for the γH2AX analyses, as the two cell lines clearly differ in cell cycle phase distribution. Other sources for oxidative damage should be explored. The IC10 are about 2.5 fold increased when lacking pol beta, it does so also when inhibiting MTH1. Even though (S)-crizotinib lowered IC10, thereby suggesting a cytotoxic nature of oxidized nucleotide incorporation from KBrO₃ treatment, it did so relatively independent of pol beta status. According to the model, however, one would expect a greater difference between the cell lines under these conditions. Therefore does (S)-crizotinib mediated increased cytotoxicity to KBrO₃ in this model really depend on MTH1 or are other processes masking the pol beta involvement? Validation by knockdown experiments and cell cycle phase considerations would be

very helpful here! γ H2AX data were added to show DNA damage accumulation. The flow cytometry pictures indicate a much higher S to G1-phase ratio in the wildtype cells. This could cause increased damage due to oxidized nucleotide incorporation by replication in the S-phase cells and therefore result in higher γ H2AX. Even though the model predicts replication problems (and BER may have a role here too) this cell cycle phase distribution difference could be confounding as it may provoke changes unrelated to pol beta status or BER activity. This reviewer suggests complemented cells and/or consideration of cell cycle phases to address this in the γ H2AX and growth delay experiments (see above).

Minor:

Fig1 and all other quantifications: Kinetics are best presented by line graphs in scatter plots. Using dotted or black reference lines will also help to reduce replications of bars/data in Figure 3 and 4.

Fig1: Please add quantification of dC:dGTP for comparison purposes (see above). Why are the repair assays comparing 8oxodGTP (200uM) against dNTPs and not dGTP at the same concentration?

Fig2: See above and please specify cell lysis procedure used for quantification in Materials and Methods.

Fig 3 to 6: Graph options as in above. Please show ligation efficiencies in Sup.

Fig7: Neither the choice of ligase nor the use of polymerase alters 5'adenylate product formation dramatically. Please show ligation and/or representative gel and elaborate on the lack of changes (or consistency) in the discussion.

Sup data: The supplementary data provide valuable controls. D265A experiment quantification is missing. Sup Fig 2: an additional evaluation at about 12mM KBrO3 in the wildtypes (causing similar cytotoxicity in wildtypes to pol beta null cells) would allow both, the assessment of the linearity and the relation to the cytotoxicity.

Lastly, (alkaline) comet assays can be a valuable tool to assess BER intermediates (compared to the more general DNA damage marker γ H2AX that labels double strand breaks and replication problems) – why was this not considered?

There are some (few) phrases and spelling errors that need attention.

October 8, 2016

Re: NCOMMS-16-17057

“Oxidized nucleotide insertion by pol β confounds ligation during base excision repair.”

Point-by-point responses to the Referees’ comments are as follows:

Referee: 1

Comments for the Authors

This manuscript examines the threat of oxidized nucleotide incorporation during BER to abortive ligation. The majority of the work is biochemical with a small contribution using cells to demonstrate that oxidized nucleotides do in fact, result in greater cytotoxicity and increased γ H2AX, dependent on pol β . The work stresses the damaging effects of incomplete BER, and how continued oxidative exposure may increase the likelihood of more serious problems such as DNA breaks. The authors present a model in which the insertion of oxidized dNTPs into DNA by pol β during BER results in the generation of 5'-adenylated DNA, DNA ligation failure, and the cytotoxic DNA breaks. The manuscript is well written and the *in vitro* experiments are nicely carried out and clearly presented. However, additional *in vitro* and cell experiments are needed to substantiate the results and the main conclusions stated in the manuscript.

Response:

We thank the Referee for these positive comments. This work is a combination of biochemical and cell-based studies, and we appreciate the recognition that the majority of the results are from biochemical experiments. We have revised the manuscript to accommodate the request for additional experiments, as summarized below.

Point 1) Figure 2 clearly shows that wild-type cells are more sensitive to KBrO₃-mediated oxidative stress than pol β null cells, and that the effect is compounded by the inhibition of MTH1. The authors conclude that increased oxidative nucleotide pools contribute to pol β-mediated incorporation of oxidized nucleotides, resulting in cellular sensitivity. This is a logical conclusion, but in Figure 7, the authors demonstrate that pol λ, the backup enzyme for pol β, behaves similarly in regards to insertion of oxidized nucleotides as well as 5'-adenylate formation. It seems that pol β null cells, not acutely knocked down for pol β, but chronically depleted, should also experience abortive ligation due to pol λ activity. If the cells in question do not express pol λ in sufficient amounts, this may be a more trivial issue to address.

Response:

We found that pol β^{-/-} cells exhibit a modest level of sensitivity to KBrO₃ treatment and this sensitivity is increased by MTH1 inhibition or MTH1 gene knockout. This is logical since there are other DNA polymerases, such as pol η, pol κ, Y-family polymerase REV1 and B-family translesion polymerase pol ζ, that can incorporate oxidized nucleotides depending on various factors (Katafuchi A. and Nohmi T. *Mutat. Res.*, 2010). The 8-oxodGTP insertion by these DNA polymerases could also lead to the ligation failure. Yet, any effect these polymerases may have on the sensitivity to KBrO₃ we observed in pol β^{-/-} cells appears to be minimal. Similarly, as pointed out by the Referee, the X-family DNA polymerase pol λ, that can back-up pol β deficiency in BER, could be another source of impaired BER due the effect of its 8-oxodGMP insertion on ligation, as shown in Fig. 6c,d. Such a potential effect of pol λ, however, appears to be minimal.

Nevertheless, the primary goal of the experiments described in this manuscript was to evaluate whether oxidized nucleotide insertion by pol β confounds the DNA ligation step of the BER pathway. The information presented in the manuscript accomplishes this goal. In addition to the *in vivo* experiments with pol β^{+/+} and pol β^{-/-} cells, the evidence comes from experiments with all possible template bases against which pol β inserts oxidized purine nucleotides in BER assays coupled with ligation by both BER ligases DNA ligase I or XRCC1/DNA ligase III complex. Further, we provided interesting information by examining pol β mutants that have

important active site alterations changing the efficiency of incorporation of 8-oxodGMP opposite template dA and dC (Batra V.K. *et al. Nat. Struct. Mol. Biol.*, 2010; Batra V.K. *et al. Proc. Natl. Acad. Sci.*, 2012; Beard W.A. *et al., Mutat. Res.*, 2010; Freudenthal B. *et al. Nature*, 2016; Miller H. *et al. Biochemistry*, 2000).

We believe that additional information involving other DNA polymerases that could be obtained by additional *in vivo* experiments, as the Referee requested, are outside scope of this current work.

However, to address the Referee's concern about the possibility of abortive ligation due to pol λ activity in pol $\beta^{-/-}$ cells, we further analyzed pol λ 8-oxodGTP insertion coupled with ligation and its comparison with that of pol β in the cell extracts from pol $\beta^{+/+}$, pol $\beta^{-/-}$, pol $\lambda^{+/+}$, pol $\lambda^{-/-}$, and pol $\beta^{-/-}$ pol $\lambda^{-/-}$ cells. We present this additional data for the Referee's consideration only in Supporting Data, and we comment on the results at the end of our point-by-point responses here (Supporting Data Figures 4-6).

Point 2) Also regarding Figure 2, a paper from this same lab shows that pol β null, pol λ null, and particularly the double mutant are markedly sensitive to hydrogen peroxide. The effects of KBrO_3 are clearly different, but the text should explain the difference in those agents.

Response:

In previous work, as noted by the Referee (Braithwaite E.K. *et al. PLOS One*, 2010), we reported that wild-type cells are less sensitive than pol β null cells to H_2O_2 treatment. Yet, in the current study, we showed that pol $\beta^{+/+}$ cells are more sensitive than pol $\beta^{-/-}$ cells to KBrO_3 . We believe the difference is likely due to the nature of oxidizing agents-mediated DNA damage in cells. H_2O_2 has been widely utilized as a representative reactive oxygen species damaging agent for studies of oxidative stress in mammalian cells and is considered to induce single-strand breaks and various forms of base damage (e.g., Dizdaroglu M. *et al., Arch. Biochem. Biophys.*, 1991). In our current study, we chose to use KBrO_3 as oxidizing agent because it is known to cause oxidative stress through formation of reactive metabolites that preferentially oxidize guanine residues to 8-oxoG (Campalans A. *et al., NAR*, 2013; Campalans A. *et al., Mol. and Cell. Biology*, 2015; Spassova M.A., *et al., Oxid. Med. Cell Longev.*, 2015). Thus,

we considered H₂O₂ as less likely than KBrO₃ to specifically oxidize guanine in the nucleotide pool.

In order to address the Referee's point, we have revised the Discussion section of the manuscript (lines 215-224) to explain the difference between our previous report and the current study, and we have added the references above to the reference list of the revised manuscript (refs 34-38).

Point 3) Figure 2C: An experiment should be included with and without (S)-crizotinib and no KBrO₃ treatment to get detect the steady-state concentration of oxidized dNTPs in the nucleotide pool in these cells and its effect on cell survival.

Response:

The Referee's point about the benefit of pool measurements in the pol β^{+/+} and pol β^{-/-} cells is logical in light of the information described in this manuscript, but experiments along these lines are underway in collaboration with others and are well beyond our reach at the moment. We included this explanation in the discussion part of revised manuscript (lines 257-260).

We present our *in vivo* data in Fig. 2, Supplementary Fig. 3 and Supplementary Fig. 4 in the revised manuscript, as explained in detail below (point 4). Also, we want to represent data for (S)-crizotinib here in a different presentation style that shows the effect in pol β^{+/+} and pol β^{-/-} cells treated with or without (S)-crizotinib. The figures below can be replaced with the ones in Fig. 2a,b in the revised manuscript, if needed.

Point 4) The Experiment in Fig. 2c should also be carried out following MTH1 knockdown or knockout to demonstrate the specificity of (S)-crizotinib and that MTH1 inhibition is responsible for the enhanced cytotoxicity.

Response:

In order to address the Referee's request for demonstration of the specificity of the effect of the MTH1 inhibitor (S)-crizotinib on cytotoxicity, we conducted an entirely new set of experiments involving MTH1 gene knockout in both of our MEF cell lines, pol $\beta^{+/+}$ and pol $\beta^{-/-}$ cells, and then conducted cell survival assays similar to those used with MTH1 inhibitor treatment. Pol $\beta^{+/+}$ MTH1 $^{-/-}$ cells were found more sensitive to KBrO₃ than pol $\beta^{-/-}$ MTH1 $^{-/-}$ (Fig. 2c,d). The results of these experiments are consistent with the KBrO₃-induced cell killing effect we observed in cells co-treated with (S)-crizotinib (Fig. 2a,b). In addition, (S)-crizotinib alone at the concentration used in the combination studies (6 μ M) had minimal effect (Supplementary Fig. 3). We also present western blot analysis (Supplementary Fig. 9) and sequence verification of the MTH1 gene deletion (Supplementary Fig. 10). We added these new results in the revised manuscript (lines 84-93). Also, the methodology used for construction of the MTH1 knockout cell lines using a CRSPR-eCas9 system is described in the revised manuscript (lines 301-347).

Point 5) Figures 3 and 4 appear to present the same data in multiple graphs. For example, Figure 4d is the combination of R283K data from 3d and R283A data from 4c. The same is true with the 4b, 3b, and 4a. This should be presented in a single graph with WT, R283K and R283A. For clarity, the graph could be a bit larger than presented now and still take up less space than 3 graphs containing duplicated information.

Response:

In order to address the Referee's point regarding representing the same data in multiple graphs, we combined the data for pol β wild-type, R283K, and R283A mutants in a single graph to compare the amount of 5'-adenylate products, (Fig. 3c,d) in the revised manuscript. Also, we present a comparison of the amounts of products for insertion as well as ligation between wild-type and pol β active site mutants (Supplementary Fig. 5).

We have revised the text accordingly in the revised manuscript (lines 107-111, 115-120). We appreciate the Referee’s assistance in clarifying these points.

Also, we included quantifications for ligation efficiencies of the data presented in the revised manuscript as listed below:

	5'-adenylation	Ligation
pol β wild-type vs R283A and R283K mutants	Figure 3c,d	Supp. Figure 5c,d
Template base effect of 8-oxodGTP on ligation failure	Figure 5a	Supp. Figure 1b
Template base effect of 8-oxodATP on ligation failure	Figure 5c	Supp. Figure 1c
Template base effect of 2-OH-dATP on ligation failure	Figure 5d	Supp. Figure 1d
pol β + DNA Lig I vs pol β + XRCC1/Lig III	Figure 6a,b	Supp. Figure 7a,b
pol β + DNA Lig I vs pol λ + DNA Ligase I	Figure 6c,d	Supp. Figure 7c,d

Point 6) The association between Supplementary Figure 3 and Figure 5 could be made more clear if the authors included “quantitated in figure 5a” or similar language. Alternatively, supplementary figure 3 could be added to the main part of the paper for more clarity.

Response:

The Referee’s point has been addressed by moving Supplementary Fig. 3, showing the effect of preformed 3'-8oxoG on ligation, to Fig. 4 in the main text of the revised manuscript. We have revised the text accordingly in the revised manuscript (lines 131-138).

Point 7) Results of OGG1, NEIL1, and NTH1 should at least be included in supplement as they further demonstrate the risk of 8-oxoG incorporation in a BER intermediate as the nick can’t be ligated or repaired.

Response:

We have added a gel image showing the results for assays of activity by OGG1, NEIL1, and NTH1 (Supplementary Fig. 8c), and we revised the manuscript accordingly (lines 177-181), to address the Referee’s point. Also, we present data showing that the DNA glycosylases used in this study were active against a substrate containing 8-oxoG (Supporting Data Fig. 1). We used purified OGG1 that had been used and reported before by our group (Sassa A. *et al.*, *J. Biol. Chem.*, 2012). This reference is added

to the reference list of the revised manuscript (ref 51). For NTH1 and NEIL1, we obtained both purified enzymes as a gift from Dr. R. S. Lloyd, whose name is now included to the acknowledgements section of the revised manuscript (line 537). We now present data verifying the enzymatic activities of NTH1 and NEIL1 (Supporting Data Fig. 1). We present these data as Supporting Data for the Referee's consideration in a separate document, and data can be moved to main part or Supplementary material of the manuscript as needed.

Point 8) Supplementary Figure 2: MTH1 inhibition by (S)-crizotinib, MTH1 knockdown, and no KBrO₃ treatment, could be included to demonstrate the biological significance of oxidized dNTPs insertion by pol β and the generation of DNA strand breaks detected by γ H2AX staining.

Response:

As mentioned above (point 4), we have developed MEF cells lines pol $\beta^{+/+}$ MTH1^{-/-} as well as pol $\beta^{-/-}$ MTH1^{-/-}, and conducted new cell survival assays with them. The results are consistent with the KBrO₃-induced cell killing effect we observed in pol $\beta^{+/+}$ cells co-treated with the MTH1 inhibitor (S)-crizotinib (Fig. 2).

Also, we conducted new γ H2AX staining experiments including an additional KBrO₃ concentration (15 mM). The results revealed increased staining with 30 mM KBrO₃ compared with 15 mM. The results are consistent with the greater cytotoxicity observed at the higher concentration in pol $\beta^{+/+}$ vs pol $\beta^{-/-}$ cells (Supplementary Fig. 4). We revised the flow cytometry figure in order to clarify the cell cycle phase distribution difference in pol $\beta^{+/+}$ and pol $\beta^{-/-}$ cells (Supplementary Fig. 4b). We also present a table that shows the proportion of cells in each phase of the cell cycle with minimal changes in cell cycle stage (Supplementary Table 3). These results are consistent with the generation of DNA strand breaks in the context of oxidized dNTPs insertion by pol β . The manuscript was revised to accommodate these points (lines 97-102).

Point 9) Supplementary Figure 4: The purpose of these experiments is not obvious. However, an experiment to test the consequence of 8-oxodGMP incorporation by pol β in the presence of one or several of these enzymes in the reaction either individually or in combination could be considered.

Response:

The purpose of these experiments was to analyze 5'- and 3'-end processing enzymes that could play a role in the repair of impaired BER after pol β 8-oxodGMP insertion followed by ligation failure. We selected APTX and FEN1 for their role in potential removal of the 5'-AMP group, and Tdp1, APE1, OGG1, NEIL1, and NTH1 for their potential role in the removal of the 3'-end 8-oxoG lesion. These enzymes are well characterized and well known end-processing enzymes. On the other hand, there are many other factors, *e.g.*, XRCC1, PARP-1, *etc.*, that could have roles in end processing.

In this study, we found that FEN1 and APTX are able to remove the 5'-adenylate group from the blocked BER intermediate with 3'-8-oxoG and 5'-AMP lesions (Supplementary Fig. 8). The role of APE1 in 3'-8-oxoG excision has been reported in other studies. Here, we provide the first report of Tdp1 as a 3'-8-oxoG end-processing enzyme, in addition to its function on mismatched 3'-end DNA. However, both of these enzymes (APE1 and Tdp1) were found to be very weak against this BER intermediate.

Referee #1 has suggested additional experiments that can mimic cellular repair of the blocked BER intermediate in the context of end processing enzymes. The Referee's point about the benefit of additional *in vitro* experiments is logical in light of the information described in the manuscript. However, we suggest that APTX and FEN1 have the primary and predominant roles in the repair of impaired BER after pol β 8-oxodGMP insertion followed by ligation failure (Supplementary Fig. 8b). Especially, the role of APTX could be critical in APTX-deficient cells, a condition related to the neurological disease known as Ataxia with Oculomotor Apraxia Type 1 (AOA1). To test of this notion, we have now performed additional *in vitro* experiments to investigate the fate of the 3'-8-oxoG and 5'-AMP-containing BER intermediate in the cell extracts from wild-type and APTX-deficient cells (Supporting Data Figures 2 and 3). The results showed that 3'-end processing for 8-oxoG removal was very

weak in both cell types (Supporting Data Fig. 2). Yet, we observed strong APTX 5'-AMP removal and FEN1 excision products in the extract from wild-type cells, while FEN1 activity was predominant in the APTX-deficient cell extract (Supporting Data Fig. 3). We present these data as Supporting Data for the Referee's consideration in a separate document, and data can be moved to main part or Supplementary material of the manuscript as needed.

Point 10) - Lane 159: "...Substrates 3 and 4..." do you mean substrates 1 and 2?

Response:

Yes, the substrates 3 and 4 should be substrates 1 and 2 as listed in Supplementary Table 1 showing all of DNA substrates used in this study. We corrected this error in the revised manuscript (line 170), and we appreciate the Referees assistance very much.

Point 11) Finally, the discussion immediately drops off without any statement of summary or significance.

Response:

We revised the discussion to accommodate the Referee's point in the revised manuscript (lines 252-260).

Authors' note in addition to responses to the points of Referee #1:

To further analyze ligation inhibition coupled with pol β oxidized nucleotide insertion, we performed new BER experiments using cell extracts from pol $\beta^{+/+}$ and pol $\beta^{-/-}$ cells (Supporting Data Figures 4 and 5). Further, to address the Referee's point that pol $\beta^{-/-}$ cells should experience abortive ligation due to pol λ activity in the cells, we conducted BER experiments using cell extracts from pol $\lambda^{+/+}$, pol $\lambda^{-/-}$, and pol $\beta^{-/-}$ pol $\lambda^{-/-}$ cells (Supporting Data Fig. 6). For this purpose, we used single-nucleotide gapped substrates opposite template base C or A, and analyzed 8-oxodGTP insertion and ligation in the cell extracts (Supporting Data Figures 5,6). In control experiments with correct dGTP (Supporting Data Fig. 4a), the products of insertion and ligation were relatively

abundant (Supporting Data Fig. 4b). For 8-oxodGTP opposite template base C or A (Supporting Data Fig. 5a), both products were weak in the extracts from pol $\beta^{-/-}$ cells compared to those from pol $\beta^{+/+}$ cells (Supporting Data Fig. 5b,c). For dA:8-oxodGTP, we observed slightly more insertion products in pol $\beta^{-/-}$ cell extracts that could represent a template base preference for DNA polymerases (compare lines 8-11 in Supporting Data Fig. 5b with 5c). However, for dC:8-oxodGTP (Supporting Data Fig. 6a), we observed significant ligation products in the extracts from pol $\lambda^{-/-}$ cells (Supporting Data Fig. 6b, lines 8-11) that could be due to pol β . Very weak ligation products were observed in the extract from pol $\beta^{-/-}$ pol $\lambda^{-/-}$ cells (Supporting Data Fig. 6b, lines 12-15) similar to those from pol $\beta^{-/-}$ cells (Supporting Data Fig. 5b, lines 8-11).

Overall, these results are consistent with our *in vitro* experiments conducted with purified enzymes pol β and DNA ligase in coupled BER assays (Fig. 1). The results also support a minimal effect of pol λ -mediated oxidized nucleotide insertion on ligation failure and impaired BER. We present these data as Supporting Data for the Referee's consideration in a separate document, and data can be moved to main part or Supplementary material of the manuscript as needed.

Referee: 2

Comments for the Authors

The manuscript "Oxidized nucleotide insertion by pol β confounds ligation during base excision" by Caglayan *et al* describes how insertion of oxidized purine nucleotides by pol β can compromise ligation in BER. This study is a follow up of the recent Nature report by the same group: "Uncovering the polymerase-induced cytotoxicity of an oxidized nucleotide", now showing the consequences of pol β executed incorporation of oxidized nucleotides in *in vitro* assays and some *in vivo* (cellular) data. The manuscript presents important findings. The authors uncover the potential challenges and pitfalls of oxidative damage repair; a process that affects aging, among others, and highlights the role of BER and pol β in this context. The data support most of the conclusions made by the authors. The data, methodology and technical execution is of high quality. Proper controls

have been applied and shown. Uncertainties have been largely addressed. Even though informative and exposing potentially important elements in repair, it is not clear however, how the mechanisms revealed in the *in vitro* assays (when only pol β and ligases are present) relate to cellular repair. This reviewer supports publication of this study, however, strongly suggests the following to elevate their findings with regards to biological significance.

Response:

We thank the Referee for these positive comments. We have conducted additional experiments and revised the manuscript to accommodate the Referee's points.

Point 1) Even though 5'-AMP formation is evident and is clearly related to oxidized nucleotide incorporation, ultimately, ligation occurs to a large part. Also, with few exceptions, 5'-adenylate products amount to about 10-20% only. How do the authors envision biological significance in the context of cellular repair that also incorporates end processing enzymes and other BER elements, notably APTX? The supplementary data show a role for APTX in 5'-AMP removal – it is not really clear how these *in vitro* assays were exactly performed, please clarify. The same authors reported compensatory mechanisms before in 2014. In detail analysis when testing oxidized nucleotide incorporation with respect to APTX's role could be a valuable addition.

Response:

We envision that ligation failure and production of the 5'-adenylated intermediate could lead to accumulation of the BER intermediate in cells, and we have added additional experiments to accommodate the Referee's point about the importance of APTX. We agree with the Referee on this point.

In our previous reports (Caglayan M. *et al.*, *Nat. Struc. Mol. Biol.*, 2014 and Caglayan M. *et al.*, *Nuc. Acids Res.*, 2015), we have shown potential roles of pol β and FEN1 in removal of the 5'-adenylated-dRP group in potential complementation of APTX deficiency. In this current manuscript, we found that APTX and FEN1 are able to remove the 5'-adenylate group from the blocked BER intermediate with 3'-8-oxoG and 5'-AMP lesions

(Supplementary Fig. 8b), while 3'-end processing enzymes, APE1, Tdp1 (Supplementary Fig. 8a), OGG1, NEIL1, and NTH1 (Supplementary Fig. 8c), had only very weak or no activity on this BER substrate.

Regarding the Referee's point that cellular repair of the blocked BER intermediate could be accomplished by repeated ligase attempts or by APTX-mediated end processing, we suggest that the 5'-end lesion, *i.e.*, 5'-AMP removal, could be especially critical in the case of APTX deficiency related to the neurological disease known as Ataxia with Oculomotor Apraxia Type 1 (AOA1). To test this notion, we have now performed additional experiments to investigate the fate of the 3'-8-oxoG and 5'-AMP-containing BER intermediate in cell extracts from wild-type and APTX-deficient cells (Supporting Data Figures 2 and 3). The results show that 3'-end processing for 8-oxoG removal is very weak in both cell types (Supporting Data Fig. 2). In contrast, both 5'-AMP APTX removal and FEN1 excision removal were observed in cell extract from wild-type cells, while FEN1 activity is predominant and abundant in the extract from APTX-deficient cells (Supporting Data Fig. 3). We present these data in Supporting Data for the Referee's consideration in a separate document, and data can be moved to main part or Supplementary material of the manuscript as needed.

Finally, to address the Referee's point regarding clarification of *in vitro* assays performed to analyze the enzymatic activity of the 5'-end and 3'-end processing enzymes, we revised the text in the Materials and Methods section of the manuscript, and this now includes separate subheadings for coupled BER assays, ligation assays, and repair assays with 3'- and 5'-end processing enzymes in more detail. (lines 276-300).

Point 2) In order to provide a role of their findings for cellular repair, the authors treated cells that express wild-type pol β or were deficient in pol β with KBrO_3 . This reviewer strongly advises to substantiate these data, so to minimize potential artifactual conclusions. Cellular models should include pol β complemented cells. This also applies for the γH2AX analyses, as the two cell lines clearly differ in cell cycle phase distribution. Other sources for oxidative damage should be explored. The IC10 are about 2.5 fold increased when lacking pol β , it does so also when inhibiting MTH1. Even though (S)-crizotinib lowered IC10, thereby suggesting a cytotoxic nature of oxidized nucleotide incorporation from KBrO_3 treatment, it did so relatively

independent of pol β status. According to the model, however, one would expect a greater difference between the cell lines under these conditions. Therefore does (S)-crizotinib mediated increased cytotoxicity to KBrO_3 in this model really depend on MTH1 or are other processes masking the pol β involvement? Validation by knockdown experiments and cell cycle phase considerations would be very helpful here!

Response:

We have conducted new MTH1 validation experiments and cell cycle analysis experiments to address the Referee's points. First, in order to address the Referee's request for further clarification of the effect of MTH1 inhibitor (S)-crizotinib on increased cytotoxicity, we have performed MTH1 gene knockout in both the MEF cell lines, pol $\beta^{+/+}$ and pol $\beta^{-/-}$. We then conducted cell survival assays with both cell lines in a fashion similar to that used with the MTH1 inhibitor. Pol $\beta^{+/+}$ MTH1 $^{-/-}$ cells were found more sensitive to KBrO_3 than pol $\beta^{-/-}$ MTH1 $^{-/-}$ (Fig. 2c,d), which is consistent with the KBrO_3 -induced cell killing effect we observed in cells co-treated with (S)-crizotinib (Fig. 2a,b). In addition, (S)-crizotinib alone at the concentration used in the combination studies (6 μM) had minimal effect (Supplementary Fig. 3). We also present western blot analysis (Supplementary Fig. 9) and sequence verification of the MTH1 gene deletion (Supplementary Fig. 10). We added these new results in the revised manuscript (lines 84-93). Also, the methodology used for construction of the MTH1 knockout cell lines using a CRISPR-eCas9 system is described in the revised manuscript (lines 301-347).

The complementation experiment mentioned by the Referee was not part of the original study design for the present study, and cannot be included in the present manuscript. The passage number for the isogenic cell lines must be similar in order to compare effects, and such complemented cell lines are not available.

However, we further analyzed 8-oxodGTP insertion coupled with ligation in the cell extracts from pol $\beta^{+/+}$, pol $\beta^{-/-}$, pol $\lambda^{+/+}$, pol $\lambda^{-/-}$, and pol $\beta^{-/-}$ pol $\lambda^{-/-}$ cells. We present this additional data for the Referee's consideration only in Supporting Data, and we comment on the results at the end of our point-by-point responses here (Supporting Data Figures 4-6).

Point 3) γ H2AX data were added to show DNA damage accumulation. The flow cytometry pictures indicate a much higher S to G1-phase ratio in the wild-type cells. This could cause increased damage due to oxidized nucleotide incorporation by replication in the S-phase cells and therefore result in higher γ H2AX. Even though the model predicts replication problems (and BER may have a role here too) this cell cycle phase distribution difference could be confounding as it may provoke changes unrelated to pol β status or BER activity. This reviewer suggests complemented cells and/or consideration of cell cycle phases to address this in the γ H2AX and growth delay experiments (see above).

Response:

We revised the flow cytometry figure in order to clarify the cell cycle phase distribution difference in pol $\beta^{+/+}$ and pol $\beta^{-/-}$ cells (Supplementary Fig. 4). We also present a table (Supplementary Table 3) that shows the proportion of cells in each phase of the cell cycle. This indicates that there is not a higher S to G1-phase ratio in the wild-type cells as the Referee was suggesting here. We revised the manuscript accordingly to accommodate these points (lines 98-102).

Minor points:

Point 4) Figure 1 and all other quantifications: Kinetics are best presented by line graphs in scatter plots. Using dotted or black reference lines will also help to reduce replications of bars/data in Figure 3 and 4.

Response:

In general, we agree with the Referee on this point about use of line graphs. However, we believe that for the current study the quantification data can be presented more clearly as bar graphs, instead of line graphs. Next, to clarify confusion regarding Figures 3 and 4, suggesting that the same data appeared in multiple graphs, we now have combined the data for pol β wild-type, R283K, and R283A mutants in a single graph to compare the products for amount of 5'-adenylation (Fig. 3c,d), ligation (Supplementary Fig. 5c,d), and insertion separately (Supplementary Fig. 5a,b). We revised the manuscript accordingly to accommodate these points (lines 107-111, 115-120).

Point 5) Figure 1: Please add quantification of dC:dGTP for comparison purposes (see above).

Response:

To address the Referee's point, we quantified the data for dC:dGTP (Supplementary Fig. 1a) for comparison with dC:8-oxodGTP (Fig. 1c) for which we showed the representative gel image (Fig. 1b). We have revised the manuscript text accordingly (lines 67-68).

Point 6) Why are the repair assays comparing 8-oxodGTP (200 μ M) against dNTPs and not dGTP at the same concentration?

Response:

In the coupled BER assays, we used the same concentration (200 μ M) of an oxidized nucleotide or a correct base nucleotide (not a dNTP mixture). We selected the correct base to compare its insertion with 8-oxodGTP based on the identity of template base in single-nucleotide gapped DNA against which pol β conducts base insertion. For example, we used 8-oxodGTP or dGTP to measure insertion coupled with ligation opposite template base cytosine (as indicated at the top panel of the gel image presented in the original submission). In order to clarify this point, we revised the text in the Materials and Methods section as follows: The reaction mixture in a final volume of 10 μ l contained 50 mM Tris-HCl (pH 7.5), 100 mM KCl, 10 mM MgCl₂, 1 mM ATP, 100 μ g/ml BSA, 10% glycerol, 1 mM DTT, and 200 μ M an oxidized base nucleotide [8-oxodGTP (Jena Bioscience), 2-OH-dATP (Jena Bioscience) or 8-oxodATP (TriLink Biotechnologies)] or a normal base nucleotide [dGTP, dTTP, dATP, or dTTP (New England Biolabs)] depending on the identity of template base in DNA. We revised the manuscript accordingly to accommodate this point (lines 267-272).

Point 7) Figure 2: See above and please specify cell lysis procedure used for quantification in Materials and Methods.

Response:

The Referee's point has been addressed by adding more explanation for the cell lysis procedure. We revised the text in the Materials and Methods section of the revised manuscript as follows: "After washing, cells were further incubated as appropriate with (S)-crizotinib until counting of cell nuclei harvested from the plates (triplicate wells for each drug concentration), following cell lysis using hypotonic solution then detergent." We revised the manuscript accordingly to accommodate this point (lines 355-358).

Point 8) Fig 3 to 6: Graph options as in above. Please show ligation efficiencies in Sup.

Figure 7: Neither the choice of ligase nor the use of polymerase alters 5'-adenylate product formation dramatically. Please show ligation and/or representative gel and elaborate on the lack of changes (or consistency) in the discussion.

Response:

We included the quantifications for ligation efficiencies in the revised manuscript as listed below. We revised the manuscript to accommodate comparisons for ligation efficiencies for the data as listed below (lines 117-120, 141-143, 151-153, 163-167). We can also provide representative gel images for each data below, and include to Supplementary Data that now already contain 10 Supplementary Figures, if needed.

	5'-adenylation	Ligation
pol β wild-type vs R283A and R283K mutants	Figure 3c,d	Supp. Figure 5c,d
Template base effect of 8-oxodGTP on ligation failure	Figure 5a	Supp. Figure 1b
Template base effect of 8-oxodATP on ligation failure	Figure 5c	Supp. Figure 1c
Template base effect of 2-OH-dATP on ligation failure	Figure 5d	Supp. Figure 1d
pol β + DNA Lig I vs pol β + XRCC1/Lig III	Figure 6a,b	Supp. Figure 7a,b
pol β + DNA Lig I vs pol λ + DNA Ligase I	Figure 6c,d	Supp. Figure 7c,d

Point 9) Sup data: The supplementary data provide valuable controls. D265A experiment quantification is missing.

Response:

To address the Referee's point, we quantified the data for D256A mutant result (Supplementary Fig. 6b) and revised the manuscript accordingly (lines 124-126). For comparison with wild-type and other pol β active site mutants (R283A and R283K) including quantifications for 8-oxodGMP insertion (Figs. 1, 3, and 5), we only presented the data for D256A 8-oxodGMP. This was similar to the weak 5'-adenylation background signal observed in the control reaction with Lig I alone.

Point 10) Sup Fig 2: an additional evaluation at about 12 mM KBrO_3 in the wild-type (causing similar cytotoxicity in wild-type to pol β null cells) would allow both, the assessment of the linearity and the relation to the cytotoxicity.

Response:

In order to address the Referee's point, we repeated γH2AX staining experiments with two KBrO_3 concentrations (15 and 30 mM). The results showed proportionality over both concentrations and this was in keeping with cytotoxicity in pol $\beta^{+/+}$ vs pol $\beta^{-/-}$ cells. We replaced our previous γH2AX staining results with this new experiment and have the results as Supplementary Fig. 4b. We revised the manuscript accordingly to accommodate these points (lines 98-102).

Point 11) Lastly, (alkaline) comet assays can be a valuable tool to assess BER intermediates (compared to the more general DNA damage marker γH2AX that labels double strand breaks and replication problems) – why was this not considered?

Response:

In previous work we have both published and extensively evaluated the alkaline “comet” assay. Based on the level of variability we have found with this assay and its weak sensitivity (Masaoka, A. *et al.*, *PLOS One*, 2013; Horton J.K. *et al.*, *Cell Research*, 2008), we have decided to avoid using the assay in the future, including in the current work. The γH2AX

assay used here is far more sensitive and reproducible than the comet assay. Furthermore, the beneficial feature of using γ H2AX assay is that it can be combined with cell cycle stage assessment, as we extensively shown in this present work.

Point 12) There are some (few) phrases and spelling errors that need attention.

Response:

We corrected phrases and spelling errors in the text to accommodate the Referee's point in the revised manuscript.

We thank the Referee for these helpful comments.

Authors' note in addition to responses to the points of Referee #2:

To further analyze ligation inhibition coupled with pol β oxidized nucleotide insertion, we performed new BER experiments using cell extracts from pol $\beta^{+/+}$, pol $\beta^{-/-}$, pol $\lambda^{+/+}$, pol $\lambda^{-/-}$, and pol $\beta^{-/-}$ pol $\lambda^{-/-}$ cells (Supporting Data Figures 4 and 5). For this purpose, we used single-nucleotide gapped substrates opposite template base C or A, and analyzed 8-oxodGTP insertion and ligation in the cell extracts (Supporting Data Figures 5,6). In control experiments with correct dGTP (Supporting Data Fig. 4a), the products of insertion and ligation were relatively abundant (Supporting Data Fig. 4b). For 8-oxodGTP opposite template base C or A (Supporting Data Fig. 5a), both products were weak in the extracts from pol $\beta^{-/-}$ cells compared to those from pol $\beta^{+/+}$ cells (Supporting Data Fig. 5b,c). For dA:8-oxodGTP, we observed slightly more insertion products in pol $\beta^{-/-}$ cell extracts that could represent a template base preference for DNA polymerases (compare lines 8-11 in Supporting Data Fig. 5b with 5c). However, for dC:8-oxodGTP (Supporting Data Fig. 6a), we observed significant ligation products in the extracts from pol $\lambda^{-/-}$ cells (Supporting Data Fig. 6b, lines 8-11) that could be due to pol β . Very weak ligation products were observed in the extract from pol $\beta^{-/-}$ pol $\lambda^{-/-}$ cells (Supporting Data Fig. 6b, lines 12-15) similar to those from pol $\beta^{-/-}$ cells (Supporting Data Fig. 5b, lines 8-11).

Overall, these results are consistent with our *in vitro* experiments conducted with purified enzymes pol β and DNA ligase in coupled BER assays (Fig. 1). The results also support a minimal effect of pol λ -mediated oxidized nucleotide insertion on ligation failure and impaired BER. We present these data as Supporting Data for the Referee's consideration in a separate document, and data can be moved to main part or Supplementary material of the manuscript as needed.

The references used in the Rebuttal (in alphabetical order)

- Batra, V. K. *et al.* Mutagenic conformation of 8-oxo-7,8-dihydro-2'-dGTP in the confines of a DNA polymerase active site. *Nat. Struct. Mol. Biol.* **17**, 889-890 (2010).
- Batra, V. K., Shock, D. D., Beard, W. A., McKenna, C. A. & Wilson, S. H. Binary complex crystal structure of DNA polymerase β reveals multiple conformations of the templating 8-oxoguanine lesion. *Proc. Natl. Acad. Sci.* **109**, 113-118 (2012).
- Beard, W. A., Batra, V. K. & Wilson, S. H. DNA polymerase structure-based insight on the mutagenic properties of 8-oxoguanine. *Mutat. Res.* **703**, 18-23 (2010).
- Braithwaite, E. K. *et al.* DNA polymerases β and λ mediate overlapping and independent roles in base excision repair in mouse embryonic fibroblasts. *PLOS One* **5**:e12229 (2010).
- Freudenthal, B. D. *et al.* Uncovering the polymerase-induced cytotoxicity of an oxidized nucleotide. *Nature* **517**, 635-639 (2016).
- Çağlayan, M., Batra, V. K., Sassa, A., Prasad, R. & Wilson, S. H. Role of polymerase β in complementing aprataxin deficiency during abasic-site base excision repair. *Nat. Struct. Mol. Biol.* **21**, 497-499 (2014).
- Çağlayan, M., Horton, J. K., Prasad, R. & Wilson, S. H. Complementation of aprataxin deficiency by base excision repair enzymes. *Nucleic Acids Res.* **43**, 2271-2281 (2015).

- Campalans, A., Moritz, E., Kortulewski, T., Biard, D., Epe, B. & Radicella, J. P. Interaction with OGG1 is required for efficient recruitment of XRCC1 to base excision repair and maintenance of genic stability after exposure to oxidative stress. *Mol. Cell. Biol.* **35**: 1648-58 (2015).
- Campalans, A., Kortulewski, T., Amouroux, R., Menoni, H., Vermeulen, W. & Radicella, J. P. Distinct spatiotemporal patterns and PARP dependence of XRCC1 recruitment to single-strand break and base excision repair. *Nuc. Acids Res.* **41**: 3115-3129 (2013).
- Dizdaroglu, M., Nackerdien, Z., Chao, B. C., Gajewski, E. & Rao, G. Chemical nature of *in vivo* DNA base damage in hydrogen peroxide-treated mammalian cells. *Arch. Biochem. Biophys.* **285**: 388-390 (1991).
- Horton, J. K., Watson, M., Stefanick, D. F., Shaughnessy, D. T., Taylor, J. A. & Wilson, S. H. XRCC1 and DNA polymerase β in cellular protection against cytotoxic DNA single-strand breaks. *Cell Research*, **18**, 48-63 (2008).
- Katafuchi, A. & Nohmi, T. DNA polymerases involved in the incorporation of oxidized nucleotides into DNA: their efficiency and template base preference. *Mutat. Res.* **703**, 24-31 (2010).
- Masaoka, A., Gassman N. R., Horton J. K., Kedar, P. S., Witt, K. L., Hobbs, C. A., Kissling, G. E., Tano, K., Asagoshi, K. & Wilson, S. H. Interaction between DNA polymerase β and BRCA1. *PLOS ONE* **8**, e66801 (2013).
- Miller, H., Prasad, R., Wilson, S. H., Johnson, F. & Grollman, A. P. 8-oxodGTP incorporation by DNA polymerase β is modified by active-site residue Asn²⁷⁹. *Biochemistry* **39**, 1029-1033 (2000).

- Sassa, A. Beard, W. A., Prasad, R. & Wilson, S. H. DNA sequence context effects on the glycosylase activity of human 8-oxoguanine DNA glycosylase. *J. Biol. Chem.* **287**: 36702-36710 (2012).
- Spassova, M. A., Miller, D. J. & Nikolov, A. S. Kinetic modelling reveals the roles of reactive oxygen species scavenging and DNA repair processes in shaping the dose-response curve of KBrO₃-induced DNA damage. *Oxidative Medicine and Cellular Longevity* (2015) <http://dx.doi.org/10.1155/2015/764375>

Reviewers' Comments:

Reviewer #1 (Remarks to the Author)

Reviewer #2 (Remarks to the Author)

This reviewer's concerns were largely and appropriately addressed. Only minor errors/comments (see below) should be considered at this stage.

The authors addressed the inconsistency of some of the cellular data very nicely with additional cellular data by targeting MTH1. However, the data still do not show a large impact. There are no differential effects of S-crizotinib or just little / partly (dose-dependent) differential effects by MTH1 knockout in the polb +/+ or polb -/-. This data should have reflected the proposed effects on cytotoxicity but, somewhat surprisingly, increased cytotoxicity (by MTH1) is only visible at higher KrBrO3 doses (or higher cytotoxicity levels) in the polb+/+. A clear impact is, however, not required as it highlights the many backup opportunities present in the cell.

Minor points and figure suggestions:

- With respect to the figure choices it would be best to exchange Figure 2a and b with those presented in the response to reviewer 1.
- Model best to be placed back as main figure
- Figure 2: The abstract states that "consistent with this result, we observe cytotoxicity in oxidizing agent-treated pol β -expressing mouse fibroblasts, and enhanced cytotoxicity following MTH1 knockout or co-treatment with a MTH1 inhibitor". This reviewer acknowledges the resistance to KBrO3 when lacking polb, but a close look shows that the survival of polb+/+ is only marginally affected by MTH1 knockout. The survival values are similar for 10 and 20mM and only differ at the (less technically robust) higher concentrations (compare polb +/+ Fig 2a with polb+/+MTH1 curve in c). Considering the vector control in 2c, these are obviously clonal effect issues. At minimum a second vector control clone may help to underline that the procedure consistently affected KrBrO3 sensitivity.
- Figure 2c: Just to be absolutely sure the authors should check whether a labeling error occurred, as the data values and errors for the lines polb+/+ in Fig2c and for polb-/-MTH1-/- in Fig2d look absolutely identical throughout.
- One way to compare the effects of genetic alterations in a proper way is to calculate the fold-change in sensitivity by comparing the doses for a certain cytotoxicity. The dose resulting in a similar growth reduction as 10mM KrBrO3 is about 25mM for polB-/- with or without (S)-crizotinib! The two-fold reduction in dose that MTH1 knockout provides in polb -/- at 20mM is, however, not present in polb+/+ (only at higher doses). Taken these data into account, this reviewer suggests to address this apparent lack more clearly.
- This reviewer would suggest to add supporting data Figure 3 as suggested by the authors
- Representative gel images are not needed.
- P9 line 215-225: is discussion material
- Pg 14 line 360: states " knockdown" rather than knockout
- Pg 11 line 255: " 8-oxodGMP" ?
- This reviewer did not understand the argument against line graphs for kinetic analysis. A trained eye appreciates the opportunities to estimate rate differences (or delays) in the different analyzed conditions. This is difficult with the current presentation as bar graphs. Also it requires some extra effort for the reader to appreciate patterns in the data. This reviewer understands that some differences are accentuated by this presentation form, but it doesn't give their own efforts in trying to perform proper kinetic analysis justice.

- Sup Fig 4: example flow pictures – according to the quantification polb+/+ and -/- must be mislabeled.

November 15, 2016

Re: NCOMMS-16-17057A

“Oxidized nucleotide insertion by pol β confounds ligation during base excision repair.”

Point-by-point responses to the Referee #2’s comments are as follows:

Overall Comments for the Authors

Comment: This reviewer’s concerns were largely and appropriately addressed. Only minor errors/comments (see below) should be considered at this stage. The authors addressed the inconsistency of some of the cellular data very nicely with additional cellular data by targeting MTH1.

Response: We appreciate these positive comments and the Referee’s assistance very much.

Comment: However, the data still do not show a large impact. There are no differential effects of S-crizotinib or just little / partly (dose-dependent) differential effects by MTH knockout in the pol $\beta^{+/+}$ or pol $\beta^{-/-}$. This data should have reflected the proposed effects on cytotoxicity but, somewhat surprisingly, increased cytotoxicity (by MTH1) is only visible at higher KBrO₃ doses (or higher cytotoxicity levels) in the pol $\beta^{+/+}$. A clear impact is, however, not required as it highlights the many backup opportunities present in the cell.

Response: The data (Fig. 3c,d) shown in the revised manuscript for *MTH1* gene deletion in the pol $\beta^{+/+}$ versus pol $\beta^{-/-}$ cell lines reveals a modest but significant reduction in KBrO₃ sensitivity with *MTH1* gene deletion. The reduction in KBrO₃ sensitivity with loss of pol β is larger with the MTH1 inhibitor S-crizotinib. As mentioned by the Referee, the modest effect of *MTH1* gene deletion was surprising, but the pol β -dependent effect is significant.

Minor points and figure suggestions:

Point 1) With respect to the figure choices it would be best to exchange Figure 2a and b with those presented in the response to reviewer 1.

Response: To address the Referee's point, we have replaced Figure 2a and 2b with the figures we presented in response to Referee #1 in our previous revision. The altered figure is Fig. 3 in the revised manuscript.

Point 2) Model best to be placed back as main figure.

Response: Per the Referee's suggestion, we moved the model back to the main text as Fig. 2a and then placed the important pol $\beta^{+/+}$ versus pol $\beta^{-/-}$ cell line KBrO_3 sensitivity data as Fig. 2b. We appreciate this suggestion and agree that these alterations have improved the presentation.

Point 3) Figure 2: The abstract states that "consistent with this result, we observe cytotoxicity in oxidizing agent-treated pol β -expressing mouse fibroblasts, and enhanced cytotoxicity following MTH1 knockout or co-treatment with a MTH1 inhibitor". This reviewer acknowledges the resistance to KBrO_3 when lacking pol β , but a close look shows that the survival of pol $\beta^{+/+}$ is only marginally affected by MTH1 knockout. The survival values are similar for 10 and 20 mM and only differ at the (less technically robust) higher concentrations (compare pol $\beta^{+/+}$ Fig. 2a with pol $\beta^{+/+}\text{MTH1}^{-/-}$ curve in c). Considering the vector control in 2c, these are obviously clonal effect issues. At minimum a second vector control clone may help to underline that the procedure consistently affected KBrO_3 sensitivity.

Response: We altered the Abstract in response to this comment.

Our view of these MTH1 knockout data is that sensitivity to 20 mM KBrO_3 is higher in the knockout than in the vector control.

In our analysis with several vector controls, the behavior of the vector control cell lines was similar. For example, we examined two cell lines for pol $\beta^{+/+}$ vector control, named pol $\beta^{+/+}$ vector 1 and pol $\beta^{+/+}$ vector 2. Similar survival curves for both cell lines were obtained. Repeat KBrO_3 sensitivity experiments with the pol $\beta^{+/+}$ vector 1 cell line are also shown below. The survival curves for pol $\beta^{+/+}$ vector 1 from three independent experiments were similar.

Point 4) Figure 2c: Just to be absolutely sure the authors should check whether a labeling error occurred, as the data values and errors for the lines pol β^{+/+} in Fig 2c and for pol β^{-/-}MTH1^{-/-} in Fig 2d look absolutely identical throughout.

Response: To address the Referee's point here, we examined these data carefully. There was no error in the designation for pol β^{+/+} in Fig. 2c and pol β^{-/-}MTH1^{-/-} in Fig. 2c (the data are now presented in Fig. 3 in the revised manuscript).

Point 5) One way to compare the effects of genetic alterations in a proper way is to calculate the fold-change in sensitivity by comparing the doses for a certain cytotoxicity. The dose resulting in a similar growth reduction as 10 mM KBrO₃ is about 25 mM for pol β^{-/-} with or without (S)-crizotonib! The two-fold reduction in dose that MTH1 knockout provides in pol β^{-/-} at 20 mM is, however, not present in pol β^{+/+} (only at higher doses). Taken these data into account, this reviewer suggests to address this apparent lack more clearly.

Response: To respond to the Referee's point, we compared the change in sensitivity to KBrO₃ at the 20 % survival level. As shown in the annotated plots below, there was a change in KBrO₃ sensitivity for pol β^{+/+} or pol β^{-/-} cells compared to those with (S)-crizotonib. Similarly, there was a change in sensitivity in both the pol β^{+/+} and pol β^{-/-} cell background with MTH1 gene

deletion. However, we believe the presentation in the current revised manuscript is preferable.

Point 6) This reviewer would suggest to add supporting data Figure 3 as suggested by the authors.

Response: The Referee's point has been addressed by moving Supporting Data Figure 3 (showing the processing of the 3'-8oxoG and 5'-AMP-containing BER intermediate by cell extracts from wild-type and APTX-deficient DT40 cells) to Supplementary Fig. 9 in the revised manuscript.

Point 7) Representative gel images are not needed.

Response: To address the Referee's point, the gel images have been removed in most cases.

Point 8) Page 9 line 215-225: is discussion material

Response: We view this material as important "context" information for the experiment to be described; the material is not intended as discussion.

Point 9) Page 14 line 360: states " knockdown" rather than knockout

Response: We have corrected this error in the revised manuscript, and we appreciate the Referees assistance very much.

Point 10) Page 11 line 255: " 8-oxodGMP"?

Response: The term "8-oxodGMP" is correct.

Point 11) This reviewer did not understand the argument against line graphs for kinetic analysis. A trained eye appreciates the opportunities to estimate rate differences (or delays) in the different analyzed conditions. This is difficult with the current presentation as bar graphs. Also it requires some extra effort for the reader to appreciate patterns in the data. This reviewer understands that some differences are accentuated by this presentation form, but it doesn't give their own efforts in trying to perform proper kinetic analysis justice.

Response: To address the Referee's concern about presentation of our kinetic data, we have changed our presentation style from bar graphs to line graphs in the current revised manuscript. The presentation seems clear in this format.

Point 12) Sup Fig 4: example flow pictures – according to the quantification pol $\beta^{+/+}$ and pol $\beta^{-/-}$ must be mislabeled.

Response: To address the Referee's point here, we examined these data carefully. The mislabeling error was found only for pol $\beta^{-/-}$ cells treated with 30 mM KBrO₃. We corrected this error in flow pictures in Supplementary Figure 4 as well as quantifications in the Supplementary Table 3. We appreciate the Referees assistance very much.